# Multi-Class Uncertainty Calibration via Mutual Information Maximization-based Binning

**Kanil Patel**[1,2*], **William Beluch**[1], **Bin Yang**[2], **Michael Pfeiffer**[1], **Dan Zhang**[1]

[1]Bosch Center for Artificial Intelligence, Renningen, Germany

[2]Institute of Signal Processing and System Theory, University of Stuttgart, Stuttgart, Germany

## Abstract

Post-hoc multi-class calibration is a common approach for providing high-quality confidence estimates of deep neural network predictions. Recent work has shown that widely used scaling methods underestimate their calibration error, while alternative Histogram Binning (HB) methods often fail to preserve classification accuracy. When classes have small prior probabilities, HB also faces the issue of severe sample-inefficiency after the conversion into $K$ one-vs-rest class-wise calibration problems. The goal of this paper is to resolve the identified issues of HB in order to provide calibrated confidence estimates using only a small holdout calibration dataset for bin optimization while preserving multi-class ranking accuracy. From an information-theoretic perspective, we derive the *I-Max* concept for binning, which maximizes the mutual information between labels and quantized logits. This concept mitigates potential loss in ranking performance due to lossy quantization, and by disentangling the optimization of bin edges and representatives allows simultaneous improvement of ranking and calibration performance. To improve the sample efficiency and estimates from a small calibration set, we propose a *shared class-wise* (sCW) calibration strategy, sharing one calibrator among similar classes (e.g., with similar class priors) so that the training sets of their class-wise calibration problems can be merged to train the single calibrator. The combination of sCW and I-Max binning outperforms the state of the art calibration methods on various evaluation metrics across different benchmark datasets and models, using a small calibration set (e.g., 1k samples for ImageNet).

## 1 Introduction

Despite great ability in learning discriminative features, deep neural network (DNN) classifiers often make over-confident predictions. This can lead to potentially catastrophic consequences in safety critical applications, e.g., medical diagnosis and autonomous driving perception tasks. A multi-class classifier is perfectly calibrated if among the cases receiving the prediction distribution $\mathbf{q}$, the ground truth class distribution is also $\mathbf{q}$. The mismatch between the prediction and ground truth distribution can be measured using the Expected Calibration Error (ECE) (Guo et al., 2017; Kull et al., 2019).

Since the pioneering work of (Guo et al., 2017), scaling methods have been widely acknowledged as an efficient post-hoc multi-class calibration solution for modern DNNs. The common practice of evaluating their ECE resorts to histogram density estimation (HDE) for modeling the distribution of the predictions. However, Vaicenavicius et al. (2019) proved that with a fixed number of evaluation bins the ECE of scaling methods is underestimated even with an infinite number of samples. Widmann et al. (2019); Kumar et al. (2019); Wenger et al. (2020) also empirically showed this underestimation phenomena. This deems scaling methods as unreliable calibration solutions, as their true ECEs can be larger than evaluated, putting many applications at risk. Additionally, setting HDE also faces the bias/variance trade-off. Increasing its number of evaluation bins reduces the bias, as the evaluation quantization error is smaller, however, the estimation of the ground truth correctness begins to suffer from high variance. Fig. 1-a) shows that the empirical ECE estimates of both the raw network outputs and the temperature scaling method (TS) (Guo et al., 2017) are sensitive to the number of evaluation

---

*firstname.lastname@de.bosch.com

Code available at `https://github.com/boschresearch/imax-calibration`

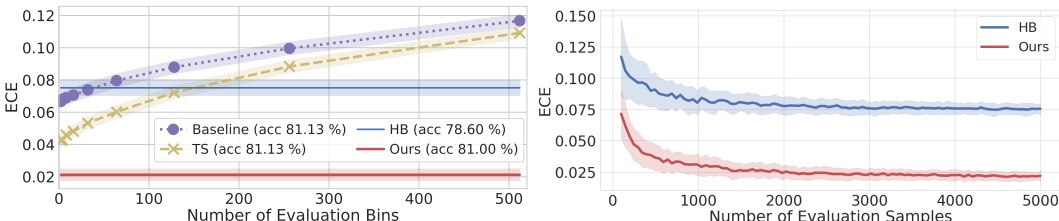

(a) Top-1 prediction ECE (5k evaluation samples)    (b) ECE converging curve (based on $10^2$ bootstraps)

Figure 1: (a) Temperature scaling (TS), equally sized-histogram binning (HB), and our proposal, i.e., sCW I-Max binning are compared for post-hoc calibrating a CIFAR100 (WRN) classifier. (b) Binning offers a reliable ECE measure as the number of evaluation samples increases.

bins. It remains unclear how to optimally choose the number of evaluation bins so as to minimize the estimation error. Recent work (Zhang et al., 2020; Widmann et al., 2019) suggested kernel density estimation (KDE) instead of HDE. However, the choice of the kernel and bandwidth also remains unclear, and the smoothness of the ground truth distribution is hard to verify in practice.

An alternative technique for post-hoc calibration is Histogram Binning (HB) (Zadrozny & Elkan, 2001; Guo et al., 2017; Kumar et al., 2019). Note, here HB is a calibration method and is different to the HDE used for evaluating ECEs of scaling methods. HB produces discrete predictions, whose probability mass functions can be empirically estimated without using HDE/KDE. Therefore, its ECE estimate is constant and unaffected by the number of evaluation bins in Fig. 1-a) and it can converge to the true value with increasing evaluation samples (Vaicenavicius et al., 2019), see Fig. 1-b).

The most common variants of HB are Equal (Eq.) size (uniformly partitioning the probability interval $[0, 1]$), and Eq. mass (uniformly distributing samples over bins) binning. These simple methods for multi-class calibration are known to degrade accuracy, since quantization through binning may remove a considerable amount of label information contained by the classifier's outputs.

In this work we show that the key for HB to retain the accuracy of trained classifiers is choosing bin edges that minimize the amount of label information loss. Both Eq. size and mass binning are suboptimal. We present *I-Max*, a novel iterative method for optimizing bin edges with proved convergence. As the location of its bin edges inherently ensures sufficient calibration samples per bin, the bin representatives of I-Max can then be effectively optimized for calibration. Two design objectives, calibration and accuracy, are thus nicely disentangled under I-Max. For multi-class calibration, I-Max adopts the one-vs-rest (OvR) strategy to individually calibrate the prediction probability of each class. To cope with a limited number of calibration samples, we propose to share one binning scheme for calibrating the prediction probabilities of similar classes, e.g., with similar class priors or belonging to the same class category. At small data regime, we can even choose to fit one binning scheme on the merged training sets of all per-class calibrations. Such a shared class-wise (sCW) calibration strategy greatly improves the sample efficiency of I-Max binning.

I-Max is evaluated according to multiple performance metrics, including accuracy, ECE, Brier and NLL, and compared against benchmark calibration methods across multiple datasets and trained classifiers. For ImageNet, I-Max obtains up to 66.11% reduction in ECE compared to the baseline and up to 38.14% reduction compared to the state-of-the-art GP-scaling method (Wenger et al., 2020).

## 2 RELATED WORK

For confidence calibration, Bayesian DNNs and their approximations, e.g. (Blundell et al., 2015) (Gal & Ghahramani, 2016) are resource-demanding methods to consider predictive model uncertainty. However, applications with limited complexity overhead and latency require sampling-free and single-model based calibration methods. Examples include modifying the training loss (Kumar et al., 2018), scalable Gaussian processes (Milios et al., 2018), sampling-free uncertainty estimation (Postels et al., 2019), data augmentation (Patel et al., 2019; Thulasidasan et al., 2019; Yun et al., 2019; Hendrycks et al., 2020) and ensemble distribution distillation (Malinin et al., 2020). In comparison, a simple approach that requires no retraining of the models is post-hoc calibration (Guo et al., 2017).

Prediction probabilities (logits) scaling and binning are the two main solutions for post-hoc calibration. Scaling methods use parametric or non-parametric models to adjust the raw logits. Guo et al. (2017) investigated linear models, ranging from the single-parameter based TS to more complicated vector/matrix scaling. To avoid overfitting, Kull et al. (2019) suggested to regularize matrix scaling with a $L_2$ loss on the model weights. Recently, Wenger et al. (2020) adopted a latent Gaussian process for multi-class calibration. Ji et al. (2019) extended TS to a bin-wise setting, by learning separate temperatures for various confidence subsets. To improve the expressive capacity of TS, an ensemble of temperatures were adopted by Zhang et al. (2020). Owing to continuous outputs of scaling methods, one critical issue discovered in the recent work is: Their empirical ECE estimate is not only non-verifiable (Kumar et al., 2019), but also asymptotically smaller than the ground truth (Vaicenavicius et al., 2019). Recent work (Zhang et al., 2020; Widmann et al., 2019) exploited KDEs for an improved ECE evaluation, however, the parameter setting requires further investigation. Nixon et al. (2019) and (Ashukha et al., 2020) discussed potential issues of the ECE metric, and the former suggested to 1) use equal mass binning for ECE evaluation; 2) measure both top-1 and class-wise ECE to evaluate multi-class calibrators, 3) only include predictions with a confidence above some epsilon in the class-wise ECE score.

As an alternative to scaling, HB quantizes the raw confidences with either Eq. size or Eq. mass bins (Zadrozny & Elkan, 2001). It offers asymptotically convergent ECE estimation (Vaicenavicius et al., 2019), but is less sample efficient than scaling methods and also suffers from accuracy loss (Guo et al., 2017). Kumar et al. (2019) proposed to perform scaling before binning for an improved sample efficiency. Isotonic regression (Zadrozny & Elkan, 2002) and Bayesian binning into quantiles (BBQ) (Naeini et al., 2015) are often viewed as binning methods. However, their ECE estimates face the same issue as scaling methods: though isotonic regression fits a piecewise linear function, its predictions are continuous as they are interpolated for unseen data. BBQ considers multiple binning schemes with different numbers of bins, and combines them using a continuous Bayesian score, resulting in continuous predictions.

In this work, we improve the current HB design by casting bin optimization into a MI maximization problem. Furthermore, our findings can also be used to improve scaling methods.

## 3 METHOD

Here we introduce the I-Max binning scheme, which addresses the issues of HB in terms of preserving label-information in multi-class calibration. After the problem setup in Sec. 3.1, Sec. 3.2) presents a sample-efficient technique for one-vs-rest calibration. In Sec. 3.3 we formulate the training objective of binning as MI maximization and derive a simple algorithm for I-Max binning.

### 3.1 PROBLEM SETUP

We address supervised multi-class classification tasks, where each input $\mathbf{x} \in \mathcal{X}$ belongs to one of $K$ classes, and the ground truth labels are one-hot encoded, i.e., $\mathbf{y} = [y_1, y_2, \ldots, y_K] \in \{0, 1\}^K$. Let $f : \mathcal{X} \mapsto [0, 1]^K$ be a DNN trained using cross-entropy loss. It maps each $\mathbf{x}$ onto a probability vector $\mathbf{q} = [q_1, \ldots, q_K] \in [0, 1]^K$, which is used to rank the $K$ possible classes of the current instance, e.g., $\arg\max_k q_k$ being the top-1 ranked class. As the trained classifier tends to overfit to the cross-entropy loss rather than the accuracy (i.e., $0/1$ loss), $\mathbf{q}$ as the prediction distribution is typically poorly calibrated. A post-hoc calibrator $h$ to revise $\mathbf{q}$ can deliver an improved performance. To evaluate the calibration performance of $h \circ f$, class-wise ECE averaged over the $K$ classes is a common metric, measuring the expected deviation of the predicted per-class confidence after calibration, i.e., $h_k(\mathbf{q})$, from the ground truth probability $p(y_k = 1|h(\mathbf{q}))$:

$$\text{cwECE}(h \circ f) = \frac{1}{K} \sum_{k=1}^{K} E_{\mathbf{q}=f(\mathbf{x})} \left\{ \left| p(y_k = 1|h(\mathbf{q})) - h_k(\mathbf{q}) \right| \right\}. \tag{1}$$

When $h$ is a binning scheme, $h_k(\mathbf{q})$ is discrete and thus repetitive. We can then empirically set $p(y_k = 1|h(\mathbf{q}))$ as the frequency of label-1 samples among those receiving the same $h_k(\mathbf{q})$. On the contrary, scaling methods are continuous. It is unlikely that two samples attain the same $h_k(\mathbf{q})$, thus requiring additional quantization, i.e., applying HDE for modeling the distribution of $h_k(\mathbf{q})$, or alternatively using KDE. It is noted that ideally we should compare the whole distribution $h(\mathbf{q})$ with

the ground truth $p(\mathbf{y}|h(\mathbf{q}))$. However, neither HDE nor KDE scales well with the number of classes. Therefore, the multi-class ECE evaluation often boils down to the one-dimensional class-wise ECE as in (1) or the top-1 ECE, i.e., $E\left[|p(y_{k=\arg\max_k h_k(\mathbf{q})} = 1|h(\mathbf{q})) - \max_k h_k(\mathbf{q})|\right]$.

## 3.2 ONE-VS-REST (OvR) STRATEGY FOR MULTI-CLASS CALIBRATION

HB was initially developed for two-class calibration. When dealing with multi-class calibration, it separately calibrates the prediction probability $q_k$ of each class in a *one-vs-rest* (OvR) fashion: For any class-$k$, HB takes $y_k$ as the binary label for a two-class calibration task in which the class-1 means $y_k = 1$ and class-0 collects all other $K - 1$ classes. It then revises the prediction probability $q_k$ of $y_k = 1$ by mapping its logit $\lambda_k \triangleq \log q_k - \log(1 - q_k)$ onto a given number of bins, and reproducing it with the calibrated prediction probability. Here, we choose to bin the logit $\lambda_k$ instead of $q_k$, as the former is unbounded, i.e., $\lambda_k \in \mathbb{R}$, which eases the bin edge optimization process. Nevertheless, as $q_k$ and $\lambda_k$ have a monotonic bijective relation, binning $q_k$ and $\lambda_k$ are equivalent. We note that after $K$ class-wise calibrations we avoid the extra normalization step as in (Guo et al., 2017). After OvR marginalizes the multi-class predictive distribution, each class is treated independently (see Sec. A1).

The calibration performance of HB depends on the setting of its bin edges and representatives. From a calibration set $C = \{(\mathbf{y}, \mathbf{z})\}$, we can construct $K$ training sets, i.e., $S_k = \{(y_k, \lambda_k)\} \, \forall k$, under the one-vs-rest strategy, and then optimize the class-wise (CW) HB over each training set. As two common solutions in the literature, Eq. size and Eq. mass binning focus on bin representative optimization. Their bin edge locations, on the other hand, are either fixed (independent of the calibration set) or only ensures a balanced training sample distribution over the bins. After binning the logits in the calibration set $S_k = \{(y_k, \lambda_k)\}$, the bin representatives are set as the empirical frequencies of samples with $y_k = 1$ in each bin. To improve the sample efficiency of bin representative optimization, Kumar et al. (2019) proposed to perform scaling-based calibration before HB. Namely, after properly scaling the logits $\{\lambda_k\}$, the bin representative per bin is then set as the averaged sigmoid-response of the scaled logits in $S_k$ belonging to each bin.

However, pre-scaling does not resolve the sample inefficiency issue arising from a small class prior $p_k$. The two-class ratio in $S_k$ is $p_k : 1 - p_k$. When $p_k$ is small we will need a large calibration set $C = \{(\mathbf{y}, \mathbf{x})\}$ to collect enough class-1 samples in $S_k$ for setting the bin representatives. To address this, we propose to merge $\{S_k\}$ across similar classes and then use the merged set $S$ for HB training, yielding one binning scheme shareable to multiple per-class calibration tasks, i.e., *shared* class-wise (sCW) binning instead of CW binning respectively trained on $S_k$. In Sec. 4, we respectively experiment using a single binning schemes for all classes in the balanced multi-class setting, and sharing one binning among the classes with similar class priors in the imbalanced multi-class setting. Note, both $S_k$ and $S$ serve as empirical approximations to the inaccessible ground truth distribution $p(y_k, \lambda_k)$ for bin optimization. The former suffers from high variances, arising from insufficient samples (Fig. A1-a), while the latter is biased due to having samples drawn from the other classes (Fig. A1-b). As the calibration set size is usually small, the variance is expected to outweigh the approximation error over the bias (see an empirical analysis in Sec. A2).

## 3.3 BIN OPTIMIZATION VIA MUTUAL INFORMATION (MI) MAXIMIZATION

Binning can be viewed as a quantizer $Q$ that maps the real-valued logit $\lambda \in \mathbb{R}$ to the bin interval $m \in \{1, \ldots, M\}$ if $\lambda \in \mathcal{I}_m = [g_{m-1}, g_m)$, where $M$ is the total number of bin intervals, and the bin edges $g_m$ are sorted ($g_{m-1} < g_m$, and $g_0 = -\infty, g_M = \infty$). Any logit binned to $\mathcal{I}_m$ will be reproduced to the same bin representative $r_m$. In the context of calibration, the bin representative $r_m$ assigned to the logit $\lambda_k$ is used as the calibrated prediction probability of the class-$k$. As multiple classes can be assigned with the same bin representative, we will then encounter ties when making top-$k$ predictions based on calibrated probabilities. Therefore, binning as lossy quantization generally does not preserve the raw logit-based ranking performance, being subject to potential accuracy loss.

Unfortunately, increasing $M$ to reduce the quantization error is not a good solution here. For a given calibration set, the number of samples per bin generally reduces as $M$ increases, and a reliable frequency estimation for setting the bin representatives $\{r_m\}$ demands sufficient samples per bin.

Considering that the top-$k$ accuracy reflects how well the ground truth label can be recovered from the logits, we propose bin optimization via maximizing the MI between the quantized logits $Q(\lambda)$

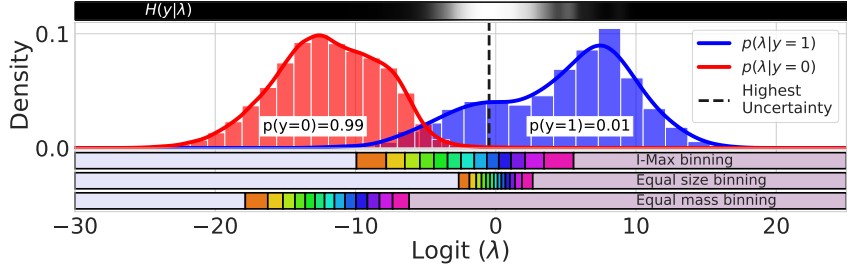

Figure 2: Histogram and KDE of CIFAR100 (WRN) logits in $S$ constructed from 1k calibration samples. The bin edges of Eq. mass binning are located at the high mass region, mainly covering class-0 due to the imbalanced two class ratio $1:99$. Both Eq. size and I-Max binning cover the high uncertainty region, but here only I-Max yields reasonable bin widths ensuring enough mass per bin. Note, Eq. size binning uniformly partitions the interval $[0,1]$ in the probability domain. The observed dense and symmetric bin location around zero is the outcome of probability-to-logit translation.

and the label $y$

$$\{g_m^*\} = \arg \max_{Q:\{g_m\}} I(y; m = Q(\lambda)) \overset{(a)}{=} \arg \max_{Q:\{g_m\}} H(m) - H(m|y) \qquad (2)$$

where the index $m$ is viewed as a discrete random variable with $P(m|y) = \int_{g_{m-1}}^{g_m} p(\lambda|y)\mathrm{d}\lambda$ and $P(m) = \int_{g_{m-1}}^{g_m} p(\lambda)\mathrm{d}\lambda$, and the equality $(a)$ is based the relation of MI to the entropy $H(m)$ and conditional entropy $H(m|y)$ of $m$. Such a formulation offers a quantizer $Q^*$ optimal at preserving the label information for a given budget on the number of bins. Unlike designing distortion-based quantizers, the reproducer values of raw logits, i.e., the bin representatives $\{r_m\}$, are not a part of the optimization space, as it is sufficient to know the mapped bin index $m$ of each logit. Once the bin edges $\{g_m^*\}$ are obtained, the bin representative $r_m$ to achieve zero calibration error shall equal $P(y = 1|m)$, which can be empirically estimated from the samples within the bin interval $\mathcal{I}_m$.

It is interesting to analyze the objective function after the equality $(a)$ in (2). The first term $H(m)$ is maximized if $P(m)$ is uniform, which is attained by Eq. mass binning. A uniform sample distribution over the bins is a sample-efficient strategy to optimize the bin representatives for the sake of calibration. However, it does not consider any label information, and thus can suffer from severe accuracy loss. Through MI maximization, we can view I-Max as revising Eq. mass by incorporating the label information into the optimization objective, i.e., having the second term $H(m|y)$. As a result, I-Max not only enjoys a well balanced sample distribution for calibration, but also maximally preserved label information for accuracy.

In the example of Fig. 2, the bin edges of I-Max binning are densely located in an area where the uncertainty of $y$ given the logit is high. This uncertainty results from small gaps between the top class predictions. With small bin widths, such nearby prediction logits are more likely located to different bins, and thus distinguishable after binning. On the other hand, Eq. mass binning has a single bin stretching across this high-uncertainty area due to an imbalanced ratio between the $p(\lambda|y = 1)$ and $p(\lambda|y = 0)$ samples. Eq. size binning follows a pattern closer to I-Max binning. However, its very narrow bin widths around zero may introduce large empirical frequency estimation errors when setting the bin representatives.

For solving the problem (2), we formulate an equivalent problem.

**Theorem 1.** *The MI maximization problem given in (2) is equivalent to*

$$\max_{Q:\{g_m\}} I(y; m = Q(\lambda)) \equiv \min_{\{g_m, \phi_m\}} \mathcal{L}(\{g_m, \phi_m\}) \qquad (3)$$

*where the loss $\mathcal{L}(\{g_m, \phi_m\})$ is defined as*

$$\mathcal{L}(\{g_m, \phi_m\}) \triangleq \sum_{m=0}^{M-1} \int_{g_m}^{g_{m+1}} p(\lambda) \sum_{y' \in \{0,1\}} P(y = y'|\lambda) \log \frac{P(y = y')}{\sigma\left[(2y' - 1)\phi_m\right]} \mathrm{d}\lambda \qquad (4)$$

*and $\{\phi_m\}$ as a set of real-valued auxiliary variables are introduced here to ease the optimization.*

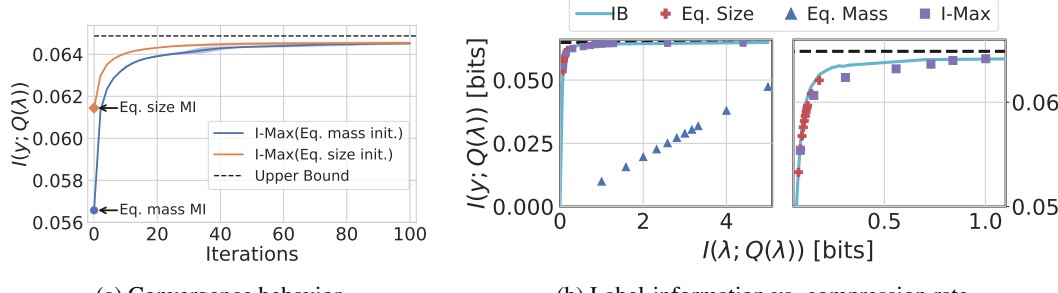

(a) Convergence behavior

(b) Label-information vs. compression rate

Figure 3: MI evaluation: The KDEs of $p(\lambda|y)$ for $y \in \{0, 1\}$ shown in Fig. 2 are used as the ground truth distribution to synthesize a dataset $S_{\text{kde}}$ and evaluate the MI of Eq. mass, Eq. size, and I-Max binning trained over $S_{\text{kde}}$. (a) The developed iterative solution for I-Max bin optimization over $S_{\text{kde}}$ successfully increases the MI over iterations, approaching the theoretical upper bound $I(y; \lambda)$. For comparison, I-Max is initialized with both Eq. size and Eq. mass bin edges, both of which are suboptimal at label information preservation. (b) We compare the three binning schemes with 2 to 16 quantization levels against the IB limit (Tishby et al., 1999) on the label-information $I(y; Q(\lambda))$ vs. the compression rate $I(\lambda; Q(\lambda))$. The information-rate pairs achieved by I-Max binning are very close to the limit. The information loss of Eq. mass binning is considerably larger, whereas Eq. size binning gets stuck in the low rate regime, failing to reach the upper bound even with more bins.

***Proof.*** See A3 for the proof. □

Next, we compute the derivatives of the loss $\mathcal{L}$ with respect to $\{g_m, \phi_m\}$. When the conditional distribution $P(y|\lambda)$ takes the sigmoid model, i.e., $P(y|\lambda) \approx \sigma[(2y - 1)\lambda]$, the stationary points of $\mathcal{L}$, zeroing the gradients over $\{g_m, \phi_m\}$, have a closed-form expression

$$g_m = \log\left\{\frac{\log\left[\frac{1+e^{\phi_m}}{1+e^{\phi_{m-1}}}\right]}{\log\left[\frac{1+e^{-\phi_{m-1}}}{1+e^{-\phi_m}}\right]}\right\}, \quad \phi_m = \log\left\{\frac{\int_{g_m}^{g_{m+1}}\sigma(\lambda)p(\lambda)\mathrm{d}\lambda}{\int_{g_m}^{g_{m+1}}\sigma(-\lambda)p(\lambda)\mathrm{d}\lambda}\right\} \approx \log\left\{\frac{\sum_{\lambda_n \in \mathcal{S}_m}\sigma(\lambda_n)}{\sum_{\lambda_n \in \mathcal{S}_m}\sigma(-\lambda_n)}\right\}, \quad (5)$$

where the approximation for $\phi_m$ arises from using the logits in the training set $S$ as an empirical approximation to $p(\lambda)$ and $S_m \overset{\Delta}{=} S \cap [g_m, g_{m+1})$. So, we can solve the problem by iteratively and alternately updating $\{g_m\}$ and $\{\phi_m\}$ based on (5) (see Algo. 1 in the appendix for pseudocode). The convergence and initialization of such an iterative method as well as the sigmoid-model assumption are discussed along with the proof of Theorem 1 in Sec. A3.

As the iterative method operates under an approximation of the inaccessible ground truth distribution $p(y, \lambda)$, we synthesize an example, see Fig. 3, to assess its effectiveness. As quantization can only reduce the MI, we evaluate $I(y; \lambda)$, serving as the upper bound in Fig. 3-a) for $I(y; Q(\lambda))$. Among the three realizations of $Q$, I-Max achieves higher MI than Eq. size and Eq. mass, and more importantly, it approaches the upper bound over the iterations. Next, we assess the performance within the framework of information bottleneck (IB) (Tishby et al., 1999), see Fig. 3-b). In the context of our problem, IB tackles $\min 1/\beta \times I(\lambda; Q(\lambda)) - I(y; Q(\lambda))$ with the weight factor $\beta > 0$ to balance between 1) maximizing the information rate $I(y; Q(\lambda))$, and 2) minimizing the compression rate $I(\lambda; Q(\lambda))$. By varying $\beta$, IB gives the maximal achievable information rate for the given compression rate. Fig. 3-b) shows that I-Max approaches the theoretical limits and provides an information-theoretic perspective on the sub-optimal performance of the alternative binning schemes. Sec. A3.2 has a more detailed discussion on the connection of IB and our problem formulation.

## 4 EXPERIMENTS

**Datasets and Models** We evaluate post-hoc calibration methods on four benchmark datasets, i.e., ImageNet (Deng et al., 2009), CIFAR 10/100 (Krizhevsky, 2009) and SVHN (Netzer et al., 2011), and across various modern DNNs architectures. More details are reported in Sec. A8.1.

**Training and Evaluation Details** We perform class-balanced random splits of the data test set, unless stated otherwise: the calibration and evaluation set sizes are both 25k for ImageNet, and 5k for

CIFAR10/100. Different to ImageNet and CIFAR10/100, the test set of SVHN is class imbalanced. We evenly split it into the calibration and evaluation set of size 13k. All reported numbers are the means across 5 random splits; stds can be found in the appendix. Note that some calibration methods only use a subset of the available calibration samples for training, showing their sample efficiency. Further calibrator training details are provided in Sec. A8.1.

We empirically evaluate MI, Accuracy (top-1 and 5 ACCs), ECE (class-wise and top-1), Brier and NLL; the latter are shown in the appendix. Analogous to (Nixon et al., 2019), we use thresholding when evaluating the class-wise ECE ($_{CW}ECE_{thr}$). Without thresholding, the empirical class-wise ECE score may be misleading. When a class-$k$ has a small class prior (e.g. $0.01$ or $0.001$), the empirical class-wise ECE score will be dominated by prediction samples where the class-$k$ is not the ground truth. For these cases, a properly trained classifier will often not rank this class-$k$ among the top classes and instead yield only small calibration errors. While it is good to have many cases with small calibration errors, they should not wash out the calibration errors of the rest of the cases (prone to poor calibration) through performance averaging. These include (1) class-$k$ is the ground truth class and not correctly ranked and (2) the classifier mis-classifies some class-$j$ as class-$k$. The thresholding remedies the washing out by focusing more on crucial cases (i.e. only averaging across cases where the prediction of the class-$k$ is above a threshold). In all experiments, our primary choice of threshold is to set it according to the class prior for the reason that the class-$k$ is unlikely to be the ground truth if its a-posteriori probability becomes lower than its prior after observing the sample.

While empirical ECE estimation of binning schemes is simple, we resort to HDE with 100 equal size evaluation bins (Wenger et al., 2020) for scaling methods. Sec. A6 also reports the results attained by HDE with additional binning schemes and KDE. For HDE-based ones, we notice that with 100 evaluation bins, the ECE estimate is insensitive to the choice of binning scheme.

### 4.1 Eq. Size, Eq. Mass vs. I-Max Binning

In Tab. 1, we compare three binning schemes: Eq. size, Eq. mass and I-Max binning. The accuracy performances of the binning schemes are proportional to their MI; Eq. mass binning is highly sub-optimal at label information preservation, and thus shows a severe accuracy drop. Eq. size binning accuracy is more similar to that of I-Max binning, but still lower, in particular at $Acc_{top5}$. Also note that I-Max approaches the MI theoretical limit of $I(y; \lambda)=0.0068$. Advantages of I-Max become even more prominent when comparing the NLLs of the binning schemes. For all ECE evalution metrics, I-Max binning improves on the baseline calibration performance, and outperforms Eq. size binning. Eq. mass binning is out of this comparison scope due to its poor accuracy deeming the method impractical. Overall, I-Max successfully mitigates the negative impact of quantization on ACCs while still providing an improved and verifiable ECE performance. Additionally, one-for-all sCW I-Max achieves an even better calibration with only 1k calibration samples, instead of the standard CW binning with 25k calibration samples, highlighting the effectiveness of the sCW strategy.

Furthermore, it is interesting to note that $_{CW}ECE$ of the Baseline classifier is very small, i.e., $0.000442$, thus it may appear as the Baseline classifier is well calibrated. However, $_{top1}ECE$ is much larger, i.e., $0.0357$. Such inconsistent observations disappear after thresholding the class-wise ECE with the class prior. This example confirms the necessity of thresholding the class-wise ECE.

In Sec. A5 we perform additional ablations on the number of bins and calibration samples. Accordingly, a post-hoc analysis investigates how the quantization error of the binning schemes change the ranking order. Observations are consistent with the intuition behind the problem formulation (see Sec. 3.3) and empirical results from Tab. 1 that MI maximization is a proper criterion for multi-class calibration and it maximally mitigates the potential accuracy loss.

### 4.2 Scaling vs. I-Max Binning

In Tab. 2, we compare I-Max binning to benchmark scaling methods. Namely, matrix scaling with $L_2$ regularization (Kull et al., 2019) has a large model capacity compared to other parametric scaling methods, while TS (Guo et al., 2017) only uses a single parameter and MnM (Zhang et al., 2020) uses three temperatures as an ensemble of TS (ETS). As a non-parametric method, GP (Wenger et al., 2020) yields state of the art calibration performance. Additional 8 scaling methods can be found in Sec. A10. Benefiting from its model capacity, matrix scaling achieves the best accuracy. I-Max

Table 1: ACCs and ECEs of Eq. mass, Eq. size and I-Max binning for the case of ImageNet (InceptionResNetV2). Due to the poor accuracy of Eq. mass binning, its ECEs are not considered for comparison. The MI is empirically evaluated based on KDE analogous to Fig. 3, where the MI upper bound is $I(y; \lambda)$=0.0068. For the other datasets and models, we refer to A9.

| Binn. | sCW(?) | size | MI ↑ | Acc$_{top1}$ ↑ | Acc$_{top5}$ ↑ | $_{CW}$ECE ↓ | $_{CW}$ECE$_{cls-prior}$ ↓ | $_{top1}$ECE ↓ | NLL ↓ |
|---|---|---|---|---|---|---|---|---|---|
| Baseline | ✗ | - | - | **80.33** | **95.10** | 0.000442 | 0.0486 | 0.0357 | 0.8406 |
| Eq. Mass | ✗ | 25k | 0.0026 | 7.78 | 27.92 | 0.000173 | 0.0016 | 0.0606 | 3.5960 |
| | ✓ | 1k | 0.0026 | 5.02 | 26.75 | 0.000165 | 0.0022 | 0.0353 | 3.5272 |
| Eq. Size | ✗ | 25k | 0.0053 | 78.52 | 89.06 | 0.000310 | 0.1344 | 0.0547 | 1.5159 |
| | ✓ | 1k | 0.0062 | 80.14 | 88.99 | 0.000298 | 0.1525 | 0.0279 | 1.2671 |
| I-Max | ✗ | 25k | **0.0066** | 80.27 | 95.01 | 0.000346 | 0.0342 | 0.0329 | 0.8499 |
| | ✓ | 1k | **0.0066** | 80.20 | 94.86 | **0.000296** | **0.0302** | **0.0200** | **0.7860** |

Table 2: ACCs and ECEs of I-Max binning (15 bins) and scaling methods. All methods use 1k calibration samples, except for Mtx. Scal. and ETS-MnM, which requires the complete calibration set, i.e., 25k/5k for ImageNet/CIFAR100. Additional 6 scaling methods can be found in A10.

| | CIFAR100 (WRN) | | | ImageNet (InceptionResNetV2) | | | |
|---|---|---|---|---|---|---|---|
| Calibrator | Acc$_{top1}$ ↑ | $_{CW}$ECE$_{cls-prior}$ ↓ | $_{top1}$ECE ↓ | Acc$_{top1}$ ↑ | Acc$_{top5}$ ↑ | $_{CW}$ECE$_{cls-prior}$ ↓ | $_{top1}$ECE ↓ |
| Baseline | 81.35 | 0.1113 | 0.0748 | 80.33 | 95.10 | 0.0486 | 0.0357 |
| Mtx Scal. w. $L_2$ | **81.44** | 0.1085 | 0.0692 | **80.78** | **95.38** | 0.0508 | 0.0282 |
| TS | 81.35 | 0.0911 | 0.0511 | 80.33 | 95.10 | 0.0559 | 0.0439 |
| GP | 81.34 | 0.1074 | 0.0358 | 80.33 | 95.11 | 0.0485 | *0.0186* |
| ETS-MnM | 81.35 | 0.0976 | 0.0451 | 80.33 | 95.10 | 0.0479 | 0.0358 |
| I-Max | 81.30 | *0.0518* | *0.0231* | 80.20 | 94.86 | *0.0302* | 0.0200 |
| I-Max w. TS | 81.34 | **0.0510** | 0.0365 | 80.20 | 94.87 | 0.0354 | 0.0402 |
| I-Max w. GP | 81.34 | 0.0559 | **0.0179** | 80.20 | 94.87 | **0.0300** | **0.0121** |

binning achieves the best calibration on CIFAR-100; on ImageNet, it has the best $_{CW}$ECE, and is similar to GP on $_{top1}$ECE. For a broader scope of comparison, we refer to Sec. A9.

To showcase the complementary nature of scaling and binning, we investigate combining binning with GP (a top performing non-parametric scaling method, though with the drawback of high complexity) and TS (a commonly used scaling method). Here, we propose to bin the raw logits and use the GP/TS scaled logits of the samples per bin for setting the bin representatives, replacing the empirical frequency estimates. As GP is then only needed at the calibration learning phase, complexity is no longer an issue. Being mutually beneficial, GP helps improving ACCs and ECEs of binning, i.e., marginal ACC drop 0.16% (0.01%) on Acc$_{top1}$ for ImageNet (CIFAR100) and 0.24% on Acc$_{top5}$ for ImageNet; and large ECE reduction 38.27% (49.78%) in $_{CW}$ECE$_{cls-prior}$ and 66.11% (76.07%) in $_{top1}$ECE of the baseline for ImageNet (CIFAR100).

### 4.3 SHARED CLASS WISE HELPS SCALING METHODS

Though without quantization loss, some scaling methods, i.e., Beta (Kull et al., 2017), Isotonic regression (Zadrozny & Elkan, 2002), and Platt scaling (Platt, 1999), even suffer from more severe accuracy degradation than I-Max binning. As they also use the one-vs-rest strategy for multi-class calibration, we find that the proposed shared CW binning strategy is beneficial for reducing their accuracy loss and improving their ECE performance, with only 1k calibration samples, see Tab. 3.

### 4.4 IMBALANCED MULTI-CLASS SETTING

Lastly, we turn our experiments to an imbalanced multi-class setting. The adopted SVHN dataset has non-uniform class priors, ranging from 6% (e.g. digit 8) to 19% (e.g. digit 0). We reproduce Tab. 2 for SVHN, yielding Tab. 4. In order to better control the bias caused by the calibration set merging in the imbalanced multi-class setting, the former one-for-all sCW strategy in the balanced multi-class setting changes to sharing I-Max among classes with similar class priors. Despite the class imbalance,

Table 3: ACCs and ECEs of scaling methods using the one-vs-rest conversion for multi-class calibration. Here we compare using 1k samples for both CW and one-for-all sCW scaling.

| Calibrator | sCW(?) | CIFAR100 (WRN) | | | ImageNet (InceptionResNetV2) | | | |
| --- | --- | --- | --- | --- | --- | --- | --- | --- |
| | | $Acc_{top1}$ ↑ | $_{CW}ECE_{cls-prior}$ ↓ | $_{top1}ECE$ ↓ | $Acc_{top1}$ | $Acc_{top5}$ ↑ | $_{CW}ECE_{cls-prior}$ ↓ | $_{top1}ECE$ ↓ |
| Baseline | - | 81.35 | 0.1113 | 0.0748 | 80.33 | 95.10 | 0.0489 | 0.0357 |
| Beta | ✗ | 81.02 | 0.1066 | 0.0638 | 77.80 | 86.83 | 0.1662 | 0.1586 |
| Beta | ✓ | **81.35** | 0.0942 | 0.0357 | **80.33** | **95.10** | 0.0625 | 0.0603 |
| I-Max w. Beta | ✓ | 81.34 | **0.0508** | **0.0161** | 80.20 | 94.87 | **0.0381** | **0.0574** |
| Isot. Reg. (IR) | ✗ | 80.62 | 0.0989 | 0.0785 | 77.82 | 88.36 | 0.1640 | 0.1255 |
| Isot. Reg. (IR) | ✓ | 81.30 | 0.0602 | 0.0257 | **80.22** | **95.05** | 0.0345 | 0.0209 |
| I-Max w. IR | ✓ | **81.34** | **0.0515** | **0.0212** | 80.20 | 94.87 | **0.0299** | **0.0170** |
| Platt Scal. | ✗ | 81.31 | 0.0923 | 0.1035 | **80.36** | 94.91 | 0.0451 | 0.0961 |
| Platt Scal. | ✓ | **81.35** | 0.0816 | 0.0462 | 80.33 | **95.10** | 0.0565 | 0.0415 |
| I-Max w. Platt | ✓ | 81.34 | **0.0511** | **0.0323** | 80.20 | 94.87 | **0.0293** | **0.0392** |

Table 4: ACCs and ECEs of I-Max binning (15 bins) and scaling methods. All methods use 1k calibration samples, except for Mtx. Scal. and ETS-MnM, which requires the complete calibration set, i.e., 13k for SVHN. Here, we also report the class-wise ECEs using four different thresholds.

| Calibrator | $Acc_{top1}$ ↑ | $_{top1}ECE$ ↓ | $_{CW}ECE_0$ ↓ | $_{CW}ECE_{\frac{1}{K}}$ ↓ | $_{CW}ECE_{cls-prior}$ ↓ | $_{CW}ECE_{\frac{1}{2}}$ ↓ |
| --- | --- | --- | --- | --- | --- | --- |
| Baseline | 97.08 | 0.0201 | 0.0052 | 0.0353 | 0.0356 | 0.0260 |
| Mtx. Scal. w. $L_2$ | **97.09** | 0.0188 | 0.0050 | 0.0346 | 0.0349 | 0.0250 |
| ETS-MnM | 97.08 | 0.0152 | 0.0054 | 0.0379 | 0.0382 | 0.0256 |
| TS | 97.08 | 0.0106 | 0.0041 | 0.0323 | 0.0327 | 0.0206 |
| GP | 97.08 | 0.0104 | 0.0043 | 0.0340 | 0.0341 | 0.0212 |
| I-Max | 96.88 | 0.0164 | 0.0043 | 0.0244 | 0.0245 | 0.0176 |
| I-Max w. TS | 97.06 | 0.0088 | 0.0025 | 0.0156 | 0.0155 | 0.0112 |
| I-Max w. GP | 97.06 | **0.0074** | **0.0024** | **0.0148** | **0.0147** | **0.0110** |

I-Max and its variants perform best compared to the other calibrators, being similar to Tab. 2. This shows that I-Max and the sCW strategy both can generalize to imbalanced multi-class setting.

In Tab. 4, we additionally evaluate the class-wise ECE at multiple threolds. We ablate various thresholds settings, namely, 1) 0 (no thresholding); 2) the class prior; 3) $1/K$ (any class with prediction probability below $1/K$ will not be the top-1); and 4) a relatively large number $0.5$ (the case when the confidence on class-$k$ outweighs NOT class-$k$). We observe that I-Max and its variants are consistently top performing across the different thresholds.

## 5 CONCLUSION

We proposed I-Max binning for multi-class calibration, which maximally preserves the label-information under quantization, reducing potential accuracy losses. Using the shared class-wise (sCW) strategy, we also addressed the sample-inefficiency issue of binning and scaling methods that rely on one-vs-rest (OvR) for multi-class calibration. Our experiments showed that I-Max yields consistent class-wise and top-1 calibration improvements over multiple datasets and model architectures, outperforming HB and state-of-the-art scaling methods. Combining I-Max with scaling methods offers further calibration performance gains, and more importantly, ECE estimates that can converge to the ground truth in the large sample limit.

Future work will investigate extensions of I-Max that jointly calibrate multiple classes, and thereby directly model class correlations. Interestingly, even on datasets such as ImageNet which contain several closely related classes, there is no clear evidence that methods that do model class correlations, e.g. Mtrx. Scal. capture uncertainties better. In fact, I-Max empirically outperforms such methods, although all classes are calibrated independently under the OvR assumption. Non-OvR based methods may fail due to various reasons such as under-parameterized models (e.g. TS), limited data (e.g. Mtrx. Scal.) or complexity constraints (e.g. GP). Joint class calibration therefore strongly relies on new sample efficient evaluation measures that estimate how accurately class correlations are modeled, and which can be included as additional optimization criteria.

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

This document supplements the presentation of *Multi-Class Uncertainty Calibration via Mutual Information Maximization-based Binning* in the main paper with the following:

A1: No extra normalization after $K$ class-wise calibrations;

A2: $S$ vs. $S_k$ as empirical approximations to $p(\lambda_k, y_k)$ for bin optimization in Sec. 3.2;

A3: Mathematical proof for Theorem 1 and algorithm details of I-Max in Sec. 3.3;

A4: Post-hoc analysis on the experiment results in Sec. 4.1;

A5: Ablation on the number of bins and calibration set size;

A6: Empirical ECE estimation of scaling methods under multiple evaluation schemes;

A7: Post-hoc Calibration and During-Training Calibration;

A8: Training details;

A9: Extend Tab. 1 in Sec. 4.1 for more datasets and models;

A10: Extend Tab. 2 in Sec. 4.2 for more scaling methods, datasets and models.

## A1    No Extra Normalization after $K$ Class-wise Calibrations

There is a group of calibration schemes that rely on one-vs-rest conversion to turn multi-class calibration into $K$ class-wise calibrations, e.g., histogram binning (HB), Platt scaling and Isotonic regression. After per-class calibration, the calibrated prediction probabilities of all classes no longer fulfill the constraint, i.e., $\sum_{k=1}^{K} q_k \neq 1$. An extra normalization step was taken in Guo et al. (2017) to regain the normalization constraint. Here, we note that this extra normalization is unnecessary and partially undoes the per-class calibration effect. For HB, normalization will make its outputs continuous like any other scaling methods, thereby suffering from the same issue at ECE evaluation.

One-vs-rest strategy essentially marginalizes the multi-class predictive distribution over each class. After such marginalization, each class and its prediction probability shall be treated independently, thus no longer being constrained by the multi-class normalization constraint. This is analogous to train a CIFAR or ImageNet classifier with sigmoid rather than softmax cross entropy loss, e.g., Ryou et al. (2019). At training and test time, each class prediction probability is individually taken from the respective sigmoid-response without normalization. The class with the largest response is then top-ranked, and normalization itself has no influence on the ranking performance.

## A2    $S$ vs. $S_k$ as Empirical Approximations to $p(\lambda_k, y_k)$ for Bin Optimization

In Sec. 3.2 of the main paper, we discussed the sample inefficiency issue when there are classes with small class priors. Fig. A1-a) shows an example for ImageNet with 1k classes. The class prior for the class-394 is about $0.001$. Among the 10k calibration samples, we can only collect 10 samples with ground truth is the class-394. Estimating the bin representatives from these 10 samples is highly unreliable, resulting into poor calibration performance.

To tackle this, we proposed to merge the training sets $\{S_k\}$ across a selected set of classes (e.g., with similar class priors, belonging to the same class category or all classes) and use the merged $S$ to train a single binning scheme for calibrating these classes, i.e., shared class-wise (sCW) instead of CW binning. Fig. A1-b) shows that after merging over the 1k ImageNet classes, the set $S$ has sufficient numbers from both the positive $y = 1$ and negative $y = 0$ class under the one-vs-rest conversion. Tab. 1 showed the benefits of sCW over CW binnings. Tab. 3 showed that our proposal sCW is also beneficial to scaling methods which use one-vs-rest for multi-class calibration.

As pointed out in Sec. 3.2, both $S$ and $S_k$ are empirical approximations to the inaccessible ground truth $p(\lambda_k, y_k)$ for bin optimization. In Fig. A2, we empirically analyze their approximation errors. From the CIFAR10 test set, we take 5k samples to approximate per-class logit distribution $p(\lambda_k | y_k = 1)$ by means of histogram density estimation, and then use it as the baseline for comparison, i.e., $\mathrm{BS}_k$ in

---

Here, we focus on $p(\lambda_k | y_k = 1)$ as its empirical estimation suffers from small class priors, being much more challenging than $p(\lambda_k | y_k = 0)$ as illustrated in Fig. A1.

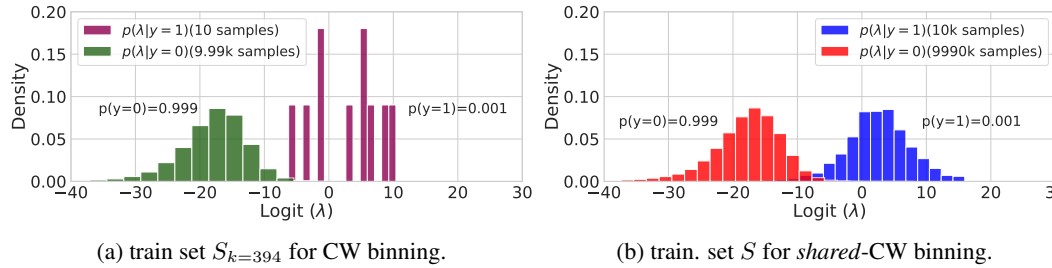

(a) train set $S_{k=394}$ for CW binning.

(b) train. set $S$ for *shared*-CW binning.

Figure A1: Histogram of ImageNet (InceptionResNetv2) logits for (a) CW and (b) sCW training. By means of the set merging strategy to handle the two-class imbalance $1:999$, $S$ has $K=1000$ times more class-1 samples than $S_k$ with the same 10k calibration samples from $C$.

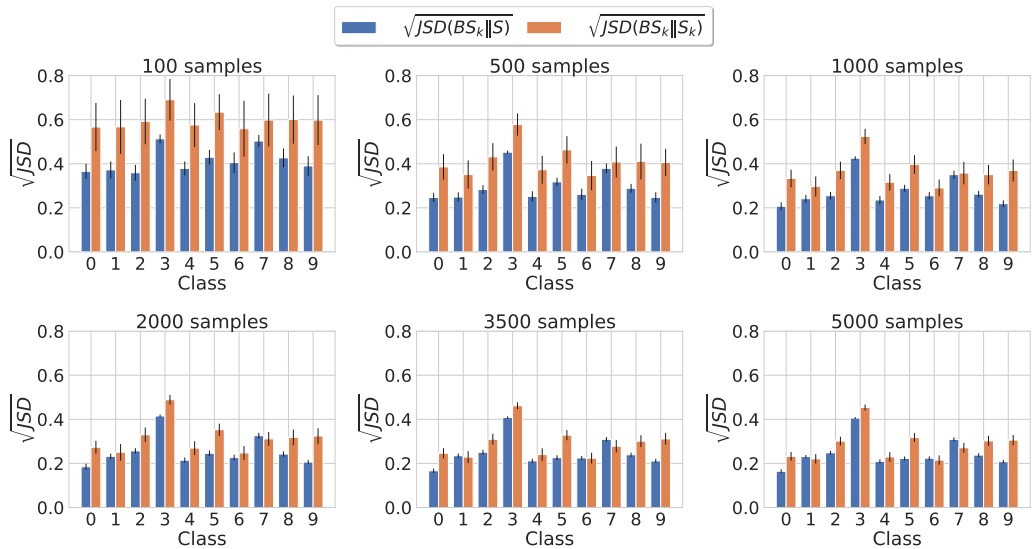

Figure A2: Empirical approximation error of $S$ vs. $S_k$, where Jensen-Shannon divergence (JSD) is used to measure the difference between the empirical distributions underlying the training sets for class-wise bin optimization. Overall, the merged set $S$ is a more sample efficient choice over $S_k$.

Fig. A2. The rest of the 5k samples in the CIFAR10 test set are reserved for constructing $S_k$ and $S$. For each class, we respectively evaluate the square root of the Jensen-Shannon divergence (JSD) from the baseline $\mathrm{BS}_k$ to the empirical distribution of $S$ or $S_k$ attained at different numbers of samples.

In general, Fig. A2 confirms that variance (due to not enough samples) outweighs bias (due to training set merging). Nevertheless, sCW does not always have smaller JSDs than CW, for instance, the class 7 with the samples larger than 2k (the blue bar "sCW" is larger than the orange bar "CW"). So, for the class-7, the bias of merging logits starts outweighing the variance when the number of samples is more than 2k. Unfortunately, we don't have more samples to further evaluate JSDs, i.e., making the variance sufficiently small to reveal the bias impact. Another reason that we don't observe large JSDs of sCW for CIFAR10 is that the logit distributions of the 10 classes are similar. Therefore, the bias of sCW is small, making CIFAR10 a good use case of sCW. From CIFAR10 to CIFAR100 and ImageNet, there are more classes with even smaller class priors. Therefore, we expect that the sample inefficiency issue of $S_k$ becomes more critical. It will be beneficial to exploit sCW for bin optimization as well as for other methods based on the one-vs-rest conversion for multi-class calibration.

---

Note, for JSD evaluation, the histogram estimator sets the bin number as the maximum of 'sturges' and 'fd' estimators, both of them optimize their bin setting towards the number of samples.

## A3   PROOF OF THEOREM 1 AND ALGORITHM DETAILS OF I-MAX

In this section, we proves Theorem 1 in Sec. 3.3, discuss the connection to the information bottleneck (IB) (Tishby et al., 1999), analyze the convergence behavior of the iterative method derived in Sec. 3.3 and modify the k-means++ algorithm (Arthur & Vassilvitskii, 2007) for initialization. To assist the implementation of the iterative method, we further provide the pseudo code and perform complexity/memory cost analysis.

### A3.1   PROOF OF THEOREM 1

**Theorem 1.** *The mutual information (MI) maximization problem given as follows:*

$$\{g_m^*\} = \arg \max_{Q:\{g_m\}} I(y; m = Q(\lambda)) \tag{A1}$$

*is equivalent to*

$$\max_{Q:\{g_m\}} I(y; m = Q(\lambda)) \equiv \min_{\{g_m, \phi_m\}} \mathcal{L}(\{g_m, \phi_m\}) \tag{A2}$$

*where the loss $\mathcal{L}(\{g_m, \phi_m\})$ is defined as*

$$\mathcal{L}(\{g_m, \phi_m\}) \triangleq \sum_{m=0}^{M-1} \int_{g_m}^{g_{m+1}} p(\lambda) \sum_{y' \in \{0,1\}} P(y = y'|\lambda) \log \frac{P(y = y')}{P_\sigma(y = y'; \phi_m)} d\lambda \tag{A3}$$

$$with \quad P_\sigma(y; \phi_m) \triangleq \sigma\left[(2y - 1)\phi_m\right]. \tag{A4}$$

*As a set of real-valued auxiliary variables, $\{\phi_m\}$ are introduced here to ease the optimization.*

*Proof.* Before staring our proof, we note that the upper-case $P$ indicates probability mass functions of discrete random variables, e.g., the label $y \in \{0, 1\}$ and the bin interval index $m \in \{1, \ldots, M\}$; whereas the lower-case $p$ is reserved for probability density functions of continuous random variables, e.g., the raw logit $\lambda \in \mathbb{R}$.

The key to prove the equivalence is to show the inequality

$$I(y; m = Q(\lambda)) \geq -\mathcal{L}(\{g_m, \phi_m\}), \tag{A5}$$

and the equality is attainable by minimizing $\mathcal{L}$ over $\{\phi_m\}$.

By the definition of MI, we firstly expand $I(y; m = Q(\lambda))$ as

$$I(y; m = Q(\lambda)) = \sum_{i=0}^{M-1} \int_{g_m}^{g_{m+1}} p(\lambda) \sum_{y' \in \{0,1\}} P(y = y'|\lambda) \log \frac{P(y = y'|m)}{P(y = y')} d\lambda, \tag{A6}$$

where the conditional distribution $P(y|m)$ is given as

$$P(y|m) = P(y|\lambda \in [g_m, g_{m+1})) = \frac{P(y) \int_{g_m}^{g_{m+1}} p(\lambda|y) d\lambda}{\int_{g_m}^{g_{m+1}} p(\lambda) d\lambda} = \frac{\int_{g_m}^{g_{m+1}} p(y|\lambda) P(y) d\lambda}{\int_{g_m}^{g_{m+1}} p(\lambda) d\lambda}. \tag{A7}$$

From the above expression, we note that MI maximization effectively only accounts to the bin edges $\{g_m\}$. The bin representatives can be arbitrary as long as they can indicate the condition $\lambda \in [g_m, g_{m+1})$. So, the bin interval index $m$ is sufficient to serve the role in conditioning the probability mass function of $y$, i.e., $P(y|m)$. After optimizing the bin edges, we have the freedom to set the bin representatives for the sake of post-hoc calibration.

Next, based on the MI expression, we compute its sum with $\mathcal{L}$

$$
\begin{aligned}
I(y; Q(\lambda)) + \mathcal{L}(\{g_m, \phi_m\}) &= \sum_{i=0}^{M-1} \int_{g_m}^{g_{m+1}} p(\lambda) \sum_{y' \in \{0,1\}} P(y = y'|\lambda) \mathrm{d}\lambda \log \frac{P(y = y'|m)}{P_\sigma(y = y'; \phi_m)} \\
&\stackrel{(a)}{=} \sum_{i=0}^{M-1} P(m) \left[ \sum_{y' \in \{0,1\}} P(y = y'|m) \log \frac{P(y = y'|m)}{P_\sigma(y = y'; \phi_m)} \right] \\
&\stackrel{(b)}{=} \sum_{i=0}^{M-1} P(m) \mathrm{KLD}\left[P(y = y'|m) \| P_\sigma(y = y'; \phi_m)\right] \\
&\stackrel{(c)}{\geq} 0.
\end{aligned}
\tag{A8}
$$

The equality $(a)$ is based on

$$
\int_{g_m}^{g_{m+1}} p(\lambda) P(y = y'|\lambda) \mathrm{d}\lambda = P(y = y', \lambda \in [g_m, g_{m+1})) = \underbrace{P(\lambda \in [g_m, g_{m+1}))}_{=P(m)} P(y = y'|m).
\tag{A9}
$$

From the equality $(a)$ to $(b)$, it is simply because of identifying the term in $[\cdot]$ of the equality $(a)$ as the Kullback-Leibler divergence (KLD) between two probability mass functions of $y$. As the probability mass function $P(m)$ and the KLD both are non-negative, we reach to the inequality at $(c)$, where the equality holds if $P_\sigma(y; \phi_m) = P(y|m)$. By further noting that $\mathcal{L}$ is convex over $\{\phi_m\}$ and $P_\sigma(y; \phi_m) = P(y|m)$ nulls out its gradient over $\{\phi_m\}$, we then reach to

$$
I(y; Q(\lambda)) + \min_{\{\phi_m\}} \mathcal{L}(\{g_m, \phi_m\}) = 0.
\tag{A10}
$$

The obtained equality then concludes our proof

$$
\begin{aligned}
\max_{\{g_m\}} I(y; Q(\lambda)) = \max_{\{g_m\}} \left[ -\min_{\{\phi_m\}} \mathcal{L}(\{g_m, \phi_m\}) \right] &= -\min_{\{g_m, \phi_m\}} \mathcal{L}(\{g_m, \phi_m\}) \\
&\equiv \min_{\{g_m, \phi_m\}} \mathcal{L}(\{g_m, \phi_m\}).
\end{aligned}
\tag{A11}
$$

$\square$

Lastly, we note that $\mathcal{L}(\{g_m, \phi_m\})$ can reduce to a NLL loss (as $P(y)$ in the log probability ratio is omittable), which is a common loss for calibrators. However, only through this equivalence proof and the MI maximization formulation, can we clearly identify the great importance of bin edges in preserving label information. So even though $\{g_m, \phi_m\}$ are jointly optimized in the equivalent problem, only $\{g_m\}$ play the determinant role in maximizing the MI.

### A3.2 CONNECTION TO INFORMATION BOTTLENECK (IB)

IB (Tishby et al., 1999) is a generic information-theoretic framework for stochastic quantization design. Viewing binning as quantization, IB aims to find a balance between two conflicting goals: 1) maximizing the information rate, i.e., the mutual information between the label and the quantized logits $I(y; Q(\lambda))$; and 2) minimizing the compression rate, i.e., mutual information between the logits and the quantized logits $I(\lambda; Q(\lambda))$. It unifies them by minimizing

$$
\min_{p(m|\lambda)} \frac{1}{\beta} I(\lambda; m = Q(\lambda)) - I(y; m = Q(\lambda)),
\tag{A12}
$$

where $m$ is the bin index assigned to $\lambda$ and $\beta$ is the weighting factor (with larger value focusing more on the information rate and smaller value on the compression rate). The compression rate is the bottleneck for maximizing the information rate. Note that IB optimizes the distribution $p(m|\lambda)$, which describes the probability of $\lambda$ being assigned to the bin with the index $m$. Since it is not a deterministic assignment, IB offers a stochastic rather than deterministic quantizer. Our information maximization formulation is a special case of IB, i.e., $\beta$ being infinitely large, as we care predominantly about how

well the label can be predicted from a compressed representation (quantized logits), in other words, making the compression rate as small as possible is not a request from the problem. For us, the only bottleneck is the number of bins usable for quantization. Furthermore, with $\beta \to \infty$, stochastic quantization degenerating to a deterministic one. If using stochastic binning for calibration, it outputs a weighted sum of all bin representatives, thereby being continuous and not ECE verifiable. Given that, we do not use it for calibration.

As the IB defines the best trade-off between the information rate and compression rate, we use it as the upper limit for assessing the optimality of I-Max in Fig. 3-b). By varying $\beta$, IB depicts the maximal achievable information rate for the given compression rate. For binning schemes (Eq. size, Eq. mass and I-Max), we vary the number of bins, and evaluate their achieved information and compression rates. As we can clearly observe from Fig. 3-b), I-Max can approach the upper limit defined by IB. Note that, the compression rate, though being measured in bits, is different to the number of bins used for the quantizer. As quantization is lossy, the compression rate defines the common information between the logits and quantized logits. The number of bins used for quantization imposes an upper limit on the information that can be preserved after quantization.

### A3.3  CONVERGENCE OF THE ITERATIVE METHOD

For convenience, we recall the update equations for $\{g_m, \phi_m\}$ in Sec. 3.3 of the main paper here

$$
\begin{aligned}
g_m &= \log \left\{ \frac{\log\left[\frac{1+e^{\phi_m}}{1+e^{\phi_{m-1}}}\right]}{\log\left[\frac{1+e^{-\phi_{m-1}}}{1+e^{-\phi_m}}\right]} \right\} & \forall m. \quad (A13) \\
\phi_m &= \log \left\{ \frac{\int_{g_m}^{g_{m+1}} \sigma(\lambda)p(\lambda)\mathrm{d}\lambda}{\int_{g_m}^{g_{m+1}} \sigma(-\lambda)p(\lambda)\mathrm{d}\lambda} \right\} \approx \log \left\{ \frac{\sum_{\lambda_n \in \mathcal{S}_m} \sigma(\lambda_n)}{\sum_{\lambda_n \in \mathcal{S}_m} \sigma(-\lambda_n)} \right\}
\end{aligned}
$$

In the following, we show that the updates on $\{g_m\}$ and $\{\phi_m\}$ according to (A13) continuously decrease the loss $\mathcal{L}$, i.e.,

$$
\mathcal{L}(\{g_m^l, \phi_m^l\}) \geq \mathcal{L}(\{g_m^{l+1}, \phi_m^l\}) \geq \mathcal{L}(\{g_m^{l+1}, \phi_m^{l+1}\}). \quad (A14)
$$

The second inequality is based on the explained property of $\mathcal{L}$. Namely, it is convex over $\{\phi_m\}$ and the minimum for any given $\{g_m\}$ is attained by $P_\sigma(y; \phi_m) = P(y|m)$. As $\phi_m$ is the log-probability ratio of $P_\sigma(y; \phi_m)$, we shall have

$$
\phi_m^{l+1} \leftarrow \log \frac{P(y = 1|m)}{P(y = 0|m)} \quad (A15)
$$

where $P(y = 1|m)$ in this case is induced by $\{g_m^{l+1}\}$ and $P(y|\lambda) = \sigma[(2y - 1)\lambda]$. Plugging $\{g_m^{l+1}\}$ and $P(y|\lambda) = \sigma[(2y - 1)\lambda]$ into (A7), the resulting $P(y = y'|m)$ at the iteration $l + 1$ yields the update equation of $\phi_m$ as given in (A13).

To prove the first inequality, we start from showing that $\{g_m^{l+1}\}$ is a local minimum of $\mathcal{L}(\{g_m, \phi_m^l\})$. The update equation on $\{g_m\}$ is an outcome of solving the stationary point equation of $\mathcal{L}(\{g_m, \phi_m^l\})$ over $\{g_m\}$ under the condition $p(\lambda = g_m) > 0$ for any $m$

$$
\frac{\partial \mathcal{L}(\{g_m, \phi_m^l\})}{\partial g_m} = p(\lambda = g_m) \sum_{y' \in \{0,1\}} P(y = y'|\lambda = g_m) \log \frac{P_\sigma(y = y'; \phi_m^l)}{P_\sigma(y = y'; \phi_{m-1}^l)} \overset{!}{=} 0 \quad \forall m
$$
$$
(A16)
$$

Being a stationary point is the necessary condition of local extremum when the function's first-order derivative exists at that point, i.e., first-derivative test. To further show that the local extremum is actually a local minimum, we resort to the second-derivative test, i.e., if the Hessian matrix of $\mathcal{L}(\{g_m, \phi_m^l\})$ is positive definite at the stationary point $\{g_m^{l+1}\}$. Due to $\phi_m > \phi_{m-1}$ with the monotonically increasing function sigmoid in its update equation, we have

$$
\left. \frac{\partial^2 \mathcal{L}(\{g_m, \phi_m^l\})}{\partial g_m \partial g_{m'}} \right|_{g_m = g_m^{l+1} \forall m} = 0 \quad \text{and} \quad \left. \frac{\partial^2 \mathcal{L}(\{g_m, \phi_m^l\})}{\partial^2 g_m} \right|_{g_m = g_m^{l+1} \forall m} > 0, \quad (A17)
$$

implying that all eigenvalues of the Hessian matrix are positive (equivalently, is positive definite). Therefore, $\{g_m^{l+1}\}$ as the stationary point of $\mathcal{L}(\{g_m, \phi_m^l\})$ is a local minimum.

It is important to note that from the stationary point equation (A16), $\{g_m^{l+1}\}$ as a local minimum is unique among $\{g_m\}$ with $p(\lambda = g_m) > 0$ for any $m$. In other words, the first inequality holds under the condition $p(\lambda = g_m^l) > 0$ for any $m$. Binning is a lossy data processing. In order to maximally preserve the label information, it is natural to exploit all bins in the optimization, not wasting any single bin in the area without mass, i.e., $p(\lambda = g_m) = 0$. Having said that, it is reasonable to constrain $\{g_m\}$ with $p(\lambda = g_m) > 0 \,\forall m$ over iterations, thereby concluding that the iterative method will converge to a local minimum based on the two inequalities (A14).

### A3.4 INITIALIZATION OF THE ITERATIVE METHOD

We propose to initialize the iterative method by modifying the k-means++ algorithm (Arthur & Vassilvitskii, 2007) that was developed to initialize the cluster centers for k-means clustering algorithms. It is based on the following identification

$$\mathcal{L}(\{g_m, \phi_m\}) + I(y; \lambda) = \sum_{i=0}^{M-1} \int_{g_m}^{g_{m+1}} p(\lambda) \text{KDL}\left[P(y = y'|\lambda)\|P_\sigma(y = y'; \phi_m)\right] \mathrm{d}\lambda \quad (A18)$$

$$\geq \int_{-\infty}^{\infty} p(\lambda) \min_m \text{KLD}\left[P(y = y'|\lambda)\|P_\sigma(y = y'; \phi_m)\right] \mathrm{d}\lambda$$

$$\approx \frac{1}{|S|} \sum_{\lambda_n \in S} \min_m \text{KLD}\left[P(y = y'|\lambda_n)\|P_\sigma(y = y'; \phi_m)\right]. \quad (A19)$$

As $I(y; \lambda)$ is a constant with respect to $(\{g_m, \phi_m\})$, minimizing $\mathcal{L}$ is equivalent to minimizing the term on the RHS of (A18). The last approximation is reached by turning the binning problem into a clustering problem, i.e., grouping the logit samples in the training set $S$ according to the KLD measure, where $\{\phi_m\}$ are effectively the centers of each cluster. k-means++ algorithm Arthur & Vassilvitskii (2007) initializes the cluster centers based on the Euclidean distance. In our case, we alternatively use the JSD as the distance measure to initialize $\{\phi_m\}$. Comparing with KLD, JSD is symmetric and bounded.

### A3.5 A REMARK ON THE ITERATIVE METHOD DERIVATION

The closed-form update on $\{g_m\}$ in (A13) is based on the sigmoid-model approximation, which has been validated through our empirical experiments. It is expected to work with properly trained classifiers that are not overly overfitting to the cross-entropy loss, e.g., using data augmentation and other regularization techniques at training. Nevertheless, even in corner cases that classifiers are poorly trained, the iterative method can still be operated without the sigmoid-model approximation. Namely, as shown in Fig. 2 of the main paper, we can resort to KDE for an empirical estimation of the ground truth distribution $p(\lambda|y)$. Using the KDEs, we can compute the gradient of $\mathcal{L}$ over $\{g_m\}$ and perform iterative gradient based update on $\{g_m\}$, replacing the closed-form based update. Essentially, the sigmoid-model approximation is only necessary to find the stationary points of the gradient equations, speeding up the convergence of the method. If attempting to keep the closed-form update on $\{g_m\}$, an alternative solution could be to use the KDEs for adjusting the sigmoid-model, e.g., $p(y|\lambda) \approx \sigma\left[(2y-1)(a\lambda + ab)\right]$, where $a$ and $b$ are chosen to match the KDE based approximation to $p(y|\lambda)$. After setting $a$ and $b$, they will be used as a scaling and bias term in the original closed-form update equations

$$\begin{cases} g_m = \frac{1}{a} \log\left\{ \dfrac{\log\left[\frac{1+e^{\phi_m}}{1+e^{\phi_{m-1}}}\right]}{\log\left[\frac{1+e^{-\phi_{m-1}}}{1+e^{-\phi_m}}\right]} \right\} - b \\[4mm] \phi_m = \log\left\{ \dfrac{\int_{g_m}^{g_{m+1}} \sigma(a\lambda + ab)p(\lambda)\mathrm{d}\lambda}{\int_{g_m}^{g_{m+1}} \sigma(-a\lambda - ab)p(\lambda)\mathrm{d}\lambda} \right\} \approx \log\left\{ \dfrac{\sum_{\lambda_n \in S_m} \sigma(a\lambda_n + ab)}{\sum_{\lambda_n \in S_m} \sigma(-a\lambda_n - ab)} \right\} \end{cases} \quad \forall m. \quad (A20)$$

### A3.6 COMPLEXITY AND MEMORY ANALYSIS

To ease the reproducibility of I-Max, we provide the pseudocode in Algorithm. 1. Based on it, we further analyze the complexity and memory cost of I-Max at training and test time.

We simplify this complexity analysis as our algorithm runs completely offline and is purely numpy-based. We note that despite the underlying (numpy) operations performed at each step of the

---

**Algorithm 1:** I-Max Binning Calibration

---

**Input:** Number of bins $M$, logits $\{\lambda_n\}_1^N$ and binary labels $\{y_n\}_1^N$
**Result:** bin edges $\{g_m\}_0^M$ ($g_0 = -\infty$ and $g_M = \infty$) and bin representations $\{\phi_m\}_0^{M-1}$
Initialization: $\{\phi_m\} \leftarrow$ Kmeans++($\{\lambda_n\}_1^N$, $M$) (see A3.4) ;
**for** $iteration = 1, 2, \ldots, 200$ **do**
    **for** $m = 1, 2, \ldots, M-1$ **do**
$$g_m \leftarrow \log \left\{ \frac{\log\left[\frac{1+e^{\phi_m}}{1+e^{\phi_{m-1}}}\right]}{\log\left[\frac{1+e^{-\phi_{m-1}}}{1+e^{-\phi_m}}\right]} \right\};$$
    **end**
    **for** $m = 0, 2, \ldots, M-1$ **do**
$$\mathcal{S}_m \triangleq \{\lambda_n\} \cap [g_m, g_{m+1}) ;$$
$$\phi_m \leftarrow \log \left\{ \frac{\sum_{\lambda_n \in \mathcal{S}_m} \sigma(\lambda_n)}{\sum_{\lambda_n \in \mathcal{S}_m} \sigma(-\lambda_n)} \right\};$$
    **end**
**end**

---

algorithm differs, we treat multiplication, division, logarithm and exponential functions each counting as the same unit cost and ignore the costs of the logic operations and add/subtract operators. The initialization has complexity of $\mathcal{O}(NM)$, for the one-dimensional logits. We exploit the sklearn implementation of Kmeans++ initialization initially used for Kmeans clustering, but replace the MSE with JSD in the distance measure. Following Algorithm 1, we arrive at the following complexity of $\mathcal{O}(N * M + I * (10 * M + 2 * M))$. Our python codes runs Algorithm. 1 within seconds for classifiers as large as ImageNet and performed purely in Numpy. The largest storage and memory consumption is for keeping the $N$ logits used during the I-Max learning phase.

At test time, there is negligible memory and storage constraints, as only $(2M - 1)$ floats need to be saved for the $M$ bin representatives $\{\phi_m\}_0^{M-1}$ and $M - 1$ bin edges $\{g_m\}_1^{M-1}$. The complexity at test time is merely logic operations to compute the bin assignments of each logit and can be done using numpy's efficient 'quantize' function. I-Max offers a real-time post-hoc calibrator which adds an almost-zero complexity and memory cost relative to the computations of the original classifier.

We will release our code soon.

## A4   POST-HOC ANALYSIS ON THE EXPERIMENT RESULTS IN SEC. 4.1

In Tab. 1 of Sec. 4.1, we compared three different binning schemes by measuring their ACCs and ECEs. The observation on their accuracy performance is aligned with our mutual information maximization viewpoint introduced in Sec. 3.3 and Fig. 2. Here, we re-present Fig. 2 and provide an alternative explanation to strengthen our understanding on how the location of bin edges affects the accuracy, e.g., why Eq. Size binning performed acceptable at the top-1 ACC, but failed at the top-5 ACC. Specifically, Fig. A3 shows the histograms of raw logits that are grouped based on their ranks instead of their labels as in Fig. 2. As expected, the logits with low ranks (i.e., rest below top-5 in Fig. A3) are small and thus take the left hand side of the plot, whereas the top-1 logits are mostly located on the right hand side. Besides sorting logits according to their ranks, we additionally estimate the density of the ground truth (GT) classes associated logits, i.e., GT in Fig. A3. With a properly trained classifier, the histogram of top-1 logits shall largerly overlap with the density curve GT, i.e., top-1 prediction being correct in most cases.

From the bin edge location of Eq. Mass binning, it attempts to attain small quantization errors for logits of low ranks rather than top-5. This will certainly degrade the accuracy performance after binning. On contrary, Eq. Size binning aims at small quantization error for the top-1 logits, but ignores top-5 ones. As a result, we observed its poor top-5 ACCs. I-Max binning nicely distributes its bin edges in the area where the GT logits are likely to locate, and the bin width becomes smaller in the area where the top-5 logits are close by (i.e., the overlap region between the red and blue histograms). Note that, any logit larger than zero must be top-1 ranked, as there can exist at most one

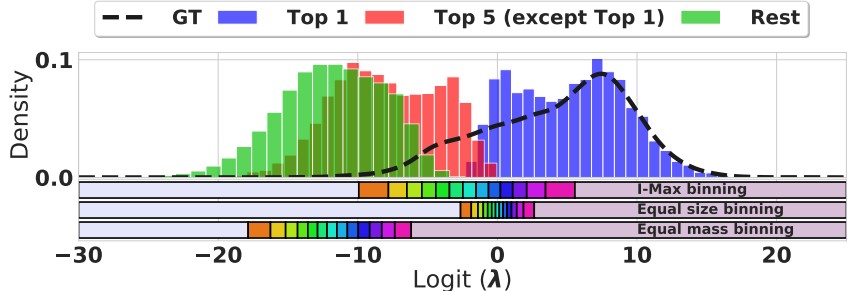

Figure A3: Histogram of CIFAR100 (WRN) logits in $S$ constructed from 1k calibration samples, using the same setting as Fig. 2 in the main paper. Instead of categorizing the logits according to their two-class label $y_k \in \{0, 1\}$ as in Fig. 2, here we sort them according to their ranks given by the CIFAR100 WRN classifier. As a baseline, we also plot the KDE of logits associated to the ground truth classes, i.e., GT.

Table A1: Comparison of sCW binning methods in the case of ImageNet - InceptionResNetV2. As sCW binning creates ties at top predictions, the ACCs initially reported in Tab. 1 of Sec. 4.1 use the class index as the secondary sorting criterion. Here, we add Acc*$_{top1}$ and Acc*$_{top5}$ which are attained by using the raw logits as the secondary sorting criterion. As the CW ECEs are not affected by this change, here we only report the new $_{top1}$ECE*.

| Binn. | Acc$_{top1}$ ↑ | Acc*$_{top1}$ ↑ | Acc$_{top5}$ ↑ | Acc*$_{top5}$ ↑ | $_{top1}$ECE ↓ | $_{top1}$ECE* ↓ | NLL ↓ |
|---|---|---|---|---|---|---|---|
| Baseline | **80.33** | - | **95.10** | - | 0.0357 | - | 0.8406 |
| Eq. Mass | 5.02 | **80.33** | 26.75 | **95.10** | 0.0353 | 0.7884 | 3.5272 |
| Eq. Size | 80.14 | 80.21 | 88.99 | **95.10** | 0.0279 | 0.0277 | 1.2671 |
| I-Max | 80.20 | **80.33** | 94.86 | **95.10** | **0.0200** | **0.0202** | **0.7860** |

class with prediction probability larger than $0.5$. Given that, the bins located above zero are no longer to maintain the ranking order, rather to reduce the precision loss of top-1 prediction probability after binning.

The second part of our post-hoc analysis is on the sCW binning strategy. When using the same binning scheme for all per-class calibration, the chance of creating ties in top-$k$ predictions is much higher than CW binning, e.g., more than one class are top-1 ranked according to the calibrated prediction probabilities. Our reported ACCs in the main paper are attained by simply returning the first found class, i.e., using the class index as the secondary sorting criterion. This is certainly a suboptimal solution. Here, we investigate on how the ties affect ACCs of sCW binning. To this end, we use raw logits (before binning) as the secondary sorting criterion. The resulting $\text{ACC}^*_{top1}$ and $\text{ACC}^*_{top5}$ are shown in Tab.A1. Interestingly, such a simple change reduces the accuracy loss of Eq. Mass and I-Max binning to zero, indicating that they can preserve the top-5 ranking order of the raw logits but not in a strict monotonic sense, i.e., some $>$ are replaced by $=$. As opposed to I-Max binning, Eq. Mass binning has a poor performance at calibration, i.e., the really high NLL and ECE. This is because it trivially ranks many classes as top-1, but each of them has a very and same small confidence score. Given that, even though the accuracy loss is no longer an issue, it is still not a good solution for multi-class calibration. For Eq. Size binning, resolving ties only helps restore the baseline top-5 but not top-1 ACC. Its poor bin representative setting due to unreliable empirical frequency estimation over too narrow bins can result in a permutation among the top-5 predictions.

Concluding from the above, our post-hoc analysis confirms that I-Max binning outperforms the other two binning schemes at mitigating the accuracy loss and multi-class calibration. In particular, there exists a simple solution to close the accuracy gap to the baseline, at the same time still retaining the desirable calibration gains.

Table A2: Ablation on the number of bins and calibration samples for sCW I-Max binning, where the basic setting is identical to the Tab. 1 in Sec. 4.1 of the main paper.

| Binn. | Bins | $Acc_{top1}\uparrow$ | $Acc_{top5}\uparrow$ | $_{CW}ECE_{\frac{1}{K}}\downarrow$ | $_{top1}ECE\downarrow$ | $Acc_{top1}\uparrow$ | $Acc_{top5}\uparrow$ | $_{CW}ECE_{\frac{1}{K}}\downarrow$ | $_{top1}ECE\downarrow$ |
|---|---|---|---|---|---|---|---|---|---|
| Baseline | - | **80.33** | 95.10 | 0.0486 | 0.0357 | **80.33** | 95.10 | 0.0486 | 0.0357 |
|  |  | 1k Calibration Samples | | | | 5k Calibration Samples | | | |
| GP |  | **80.33** | **95.11** | 0.0485 | 0.0186 | **80.33** | **95.11** | 0.0445 | 0.0177 |
| I-Max | 10 | 80.09 | 94.59 | 0.0316 | 0.0156 | 80.14 | 94.59 | 0.0330 | 0.0107 |
|  | 15 | 80.20 | 94.86 | 0.0302 | 0.0200 | 80.21 | 94.90 | 0.0257 | 0.0107 |
|  | 20 | 80.10 | 94.94 | **0.0266** | 0.0234 | 80.25 | 94.98 | **0.0220** | 0.0133 |
|  | 30 | 80.15 | 94.99 | 0.0343 | 0.0266 | 80.25 | 95.02 | 0.0310 | 0.0150 |
|  | 40 | 80.11 | 95.05 | 0.0365 | 0.0289 | 80.24 | 95.08 | 0.0374 | 0.0171 |
|  | 50 | 80.21 | 94.95 | 0.0411 | 0.0320 | 80.23 | 95.06 | 0.0378 | 0.0219 |
| I-Max w. GP | 10 | 80.09 | 94.59 | 0.0396 | 0.0122 | 80.14 | 94.59 | 0.0330 | **0.0072** |
|  | 15 | 80.20 | 94.87 | 0.0300 | **0.0121** | 80.21 | 94.88 | 0.0256 | 0.0080 |
|  | 20 | 80.23 | 94.95 | 0.0370 | 0.0133 | 80.25 | 95.00 | 0.0270 | 0.0091 |
|  | 30 | 80.26 | 95.04 | 0.0383 | 0.0141 | 80.27 | 95.02 | 0.0389 | 0.0097 |
|  | 40 | 80.27 | **95.11** | 0.0424 | 0.0145 | 80.26 | 95.08 | 0.0402 | 0.0108 |
|  | 50 | 80.30 | 95.08 | 0.0427 | 0.0153 | 80.28 | 95.08 | 0.0405 | 0.0114 |

## A5 ABLATION ON THE NUMBER OF BINS AND CALIBRATION SET SIZE

In Tab. 1 of Sec. 4.1, sCW I-Max binning is the top performing one at the ACCs, ECEs and NLL measures. In this part, we further investigate on how the number of bins and calibration set size influences its performance. Tab. A2 shows that in order to benefit from more bins we shall accordingly increase the number of calibration samples. More bins help reduce the quantization loss, but increase the empirical frequency estimation error for setting the bin representatives. Given that, we observe a reduced ACCs and increased ECEs for having 50 bins with only 1k calibration samples. By increasing the calibration set size to 5k, then we start seeing the benefits of having more bins to reduce quantization error for better ACCs. Next, we further exploit scaling method, i.e., GP Wenger et al. (2020), for improving the sample efficiency of binning at setting the bin representatives. As a result, the combination is particularly beneficial to improve the ACCs and top-1 ECE. Overall, more bins are beneficial to ACCs, while ECEs favor less number of bins.

## A6 EMPIRICAL ECE ESTIMATION OF SCALING METHODS UNDER MULTIPLE EVALUATION SCHEMES

As mentioned in the main paper, scaling methods suffer from not being able to provide verifiable ECEs, see Fig. 1. Here, we discuss alternatives to estimate their ECEs. The current literature can be split into two types of ECE evaluation: histogram density estimation (HDE) and kernel density estimation (KDE).

### A6.1 HDE-BASED ECE EVALUATION

HDE bins the prediction probabilities (logits) for density modeling. The binning scheme has different variants, where changing the bin edges can give varying measures of the ECE. Two bin edges schemes have been discussed in the literature (Eq. size and Eq. mass) as well as a new scheme was introduced (I-Max). Alternatively, we also evaluate a binning scheme which is based on KMeans clustering to determine the bin edges.

### A6.2 KDE-BASED ECE EVALUATION

Recent work (Zhang et al., 2020) presented an alternative ECE evaluation scheme which exploits KDEs to estimate the distribution of prediction probabilities $\{q_k\}$ from the test set samples. Using the code provided by Zhang et al. (2020), we observe that the KDE with the setting in their paper can

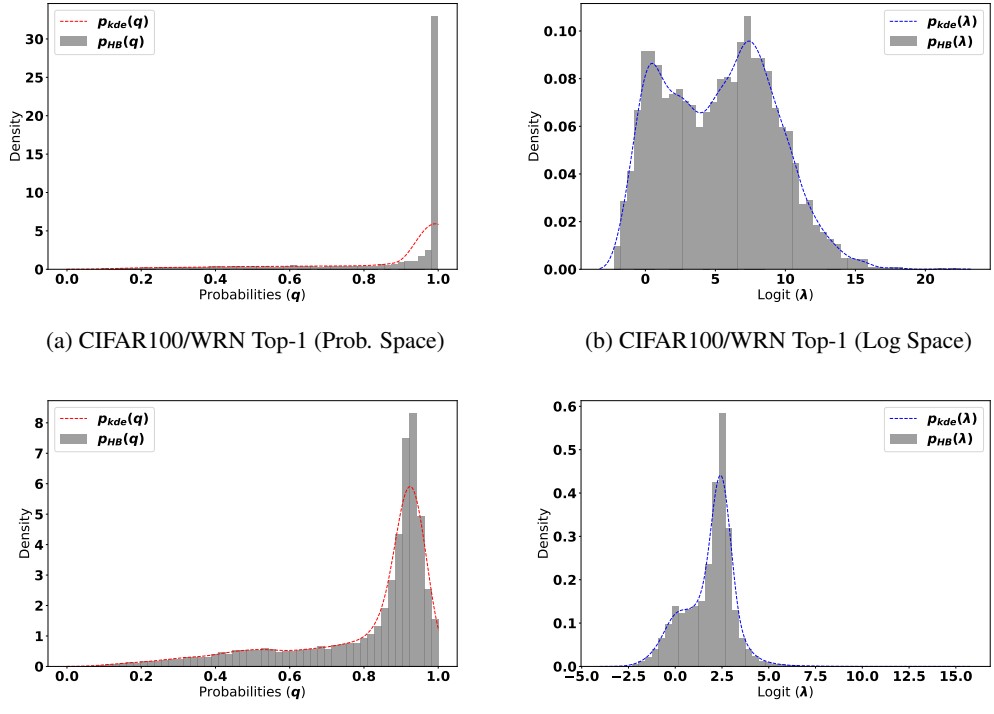

(a) CIFAR100/WRN Top-1 (Prob. Space)  (b) CIFAR100/WRN Top-1 (Log Space)

(c) ImageNet/Inceptionresnetv2 Top-1 (Prob. Space)  (d) ImageNet/Inceptionresnetv2 Top-1 (Log Space)

Figure A4: Distribution of the top-1 predictions and its log-space counterparts, i.e., $\lambda = \log q - \log(1-q)$.

have a sub-optimal fitting in the probability space. This can be observed from Fig. A4a and Fig. A4c, where the fitting is good for ImageNet/Inceptionresnetv2 though when the distribution is significantly skewed to the right (as in the case of CIFAR100/WRN) the mismatch becomes large. We expect that the case of CIFAR100/WRN is much more common in modern DNNs, due to their high capacities and prone to overfitting.

Equivalently, we can learn the distribution in its log space by the bijective transformation, i.e., $\lambda = \log q - \log(1-q)$ and $q = \sigma(\lambda)$. As we can observe from Fig. A4b and Fig. A4d, the KDE fitting for both models is consistently good.

Zhang et al. (2020) empirically validated their KDE in a toy example, where the ground truth ECE can be analytically computed. By analogy, we reproduce the experiment and further compare it with the log-space KDE evaluation. Using the same settings as in (Zhang et al., 2020), we assess the ECE evaluation error by KDE, i.e., $|\text{ECE}_{gt} - \text{ECE}_{kde}|$, in both the log and probability space, achieving prob 0.0020 vs. log 0.0017 for the toy example setting $\beta_0 = 0.5; \beta_1 = -1.5$. For an even less calibrated setting, $\beta_0 = 0.2; \beta_1 = -1.9$, we obtain prob 0.0029 vs. log 0.0020. So the log-space KDE-based ECE evaluation ($\text{kdeECE}_{log}$) has lower estimation error than in the probability space.

### A6.3 ALTERNATIVE ECE EVALUATION SCHEMES

Concluding from the above, Tab. A3 shows the ECE estimates attained by HDEs (from four different bin setting schemes) and KDE (from (Zhang et al., 2020), but in the log space). As we can see, the obtained results are evaluation scheme dependent. On contrary, I-Max binning with and without GP are not affected, and more importantly, their ECEs are better than that of scaling methods, regardless of the evaluation scheme.

Table A3: ECEs of scaling methods under various evaluation schemes for ImageNet InceptionRes-NetV2. Overall, we consider five evaluation schemes, namely (1) dECE: equal size binning; (2) mECE: equal mass binning, (3) kECE: MSE-based KMeans clustering; (4) iECE: I-Max binning; 5) kdeECE: KDE. The HDEs based schemes, i.e., (1)-(4), use $10^2$ bins. Note that, the ECEs of I-Max binning (as a calibrator rather than evaluation scheme) are agnostic to the evaluation scheme. Furthermore, BBQ suffers from severe accuracy degradation.

| Calibrator | $\text{ACC}_{\text{top}_1}$ | $_{\text{cw}}\text{dECE}_{\frac{1}{K}} \downarrow$ | $_{\text{cw}}\text{mECE}_{\frac{1}{K}} \downarrow$ | $_{\text{cw}}\text{kECE}_{\frac{1}{K}} \downarrow$ | $_{\text{cw}}\text{iECE}_{\frac{1}{K}} \downarrow$ | $_{\text{cw}}\text{kdeECE}_{\frac{1}{K}} \downarrow$ | Mean $\downarrow$ |
|---|---|---|---|---|---|---|---|
| Baseline | 80.33 | $0.0486 \pm 0.0003$ | $0.0459 \pm 0.0004$ | $0.0484 \pm 0.0004$ | $0.0521 \pm 0.0004$ | $0.0749 \pm 0.0014$ | 0.0540 |
| | | | 25k Calibration Samples | | | | |
| BBQ | 53.89 | $0.0287 \pm 0.0009$ | $0.0376 \pm 0.0014$ | $0.0372 \pm 0.0014$ | $0.0316 \pm 0.0008$ | $0.0412 \pm 0.0010$ | 0.0353 |
| Beta | 80.47 | $0.0706 \pm 0.0003$ | $0.0723 \pm 0.0005$ | $0.0742 \pm 0.0005$ | $0.0755 \pm 0.0004$ | $0.0828 \pm 0.0003$ | 0.0751 |
| Isotonic Reg. | 80.08 | $0.0644 \pm 0.0015$ | $0.0646 \pm 0.0015$ | $0.0652 \pm 0.0016$ | $0.0655 \pm 0.0015$ | $0.0704 \pm 0.0014$ | 0.0660 |
| Platt | 80.48 | $0.0597 \pm 0.0007$ | $0.0593 \pm 0.0008$ | $0.0613 \pm 0.0008$ | $0.0634 \pm 0.0008$ | $0.1372 \pm 0.0028$ | 0.0762 |
| Vec Scal. w. L2 reg. | 80.53 | $0.0494 \pm 0.0002$ | $0.0472 \pm 0.0004$ | $0.0498 \pm 0.0003$ | $0.0531 \pm 0.0003$ | $0.0805 \pm 0.0010$ | 0.0560 |
| Mtx Scal. w. L2 reg. | **80.78** | $0.0508 \pm 0.0003$ | $0.0488 \pm 0.0004$ | $0.0512 \pm 0.0005$ | $0.0544 \pm 0.0004$ | $0.0898 \pm 0.0011$ | 0.0590 |
| | | | 1k Calibration Samples | | | | |
| TS | 80.33 | $0.0559 \pm 0.0015$ | $0.0548 \pm 0.0018$ | $0.0573 \pm 0.0017$ | $0.0598 \pm 0.0015$ | $0.1003 \pm 0.0053$ | 0.0656 |
| GP | 80.33 | $0.0485 \pm 0.0037$ | $0.0450 \pm 0.0040$ | $0.0475 \pm 0.0039$ | $0.0520 \pm 0.0038$ | $0.0580 \pm 0.0052$ | 0.0502 |
| I-Max | 80.20 | | | $0.0302 \pm 0.0041$ | | | |
| I-Max w. GP | 80.20 | | | **0.0300** $\pm 0.0041$ | | | |

| Calibrator | $\text{ACC}_{\text{top}_1}$ | $_{\text{top1}}\text{dECE} \downarrow$ | $_{\text{top1}}\text{mECE} \downarrow$ | $_{\text{top1}}\text{kECE} \downarrow$ | $_{\text{top1}}\text{iECE} \downarrow$ | $_{\text{top1}}\text{kdeECE} \downarrow$ | Mean $\downarrow$ |
|---|---|---|---|---|---|---|---|
| Baseline | 80.33 | $0.0357 \pm 0.0010$ | $0.0345 \pm 0.0010$ | $0.0348 \pm 0.0012$ | $0.0352 \pm 0.0016$ | $0.0480 \pm 0.0016$ | 0.0376 |
| | | | 25k Calibration Samples | | | | |
| BBQ | 53.89 | $0.2689 \pm 0.0033$ | $0.2690 \pm 0.0034$ | $0.2690 \pm 0.0034$ | $0.2689 \pm 0.0032$ | $0.2756 \pm 0.0145$ | 0.2703 |
| Beta | 80.47 | $0.0346 \pm 0.0022$ | $0.0360 \pm 0.0017$ | $0.0360 \pm 0.0022$ | $0.0357 \pm 0.0019$ | $0.0292 \pm 0.0023$ | 0.0343 |
| Isotonic Reg. | 80.08 | $0.0468 \pm 0.0020$ | $0.0434 \pm 0.0019$ | $0.0436 \pm 0.0020$ | $0.0468 \pm 0.0015$ | $0.0437 \pm 0.0057$ | 0.0449 |
| Platt | 80.48 | $0.0775 \pm 0.0015$ | $0.0772 \pm 0.0015$ | $0.0771 \pm 0.0016$ | $0.0773 \pm 0.0014$ | $0.0772 \pm 0.0018$ | 0.0773 |
| Vec Scal. w. L2 reg. | 80.53 | $0.0300 \pm 0.0010$ | $0.0298 \pm 0.0012$ | $0.0300 \pm 0.0016$ | $0.0303 \pm 0.0011$ | $0.0365 \pm 0.0023$ | 0.0313 |
| Mtx Scal. w. L2 reg. | **80.78** | $0.0282 \pm 0.0014$ | $0.0287 \pm 0.0011$ | $0.0286 \pm 0.0014$ | $0.0289 \pm 0.0014$ | $0.0324 \pm 0.0019$ | 0.0293 |
| | | | 1k Calibration Samples | | | | |
| TS | 80.33 | $0.0439 \pm 0.0022$ | $0.0452 \pm 0.0022$ | $0.0454 \pm 0.0020$ | $0.0443 \pm 0.0020$ | $0.0679 \pm 0.0024$ | 0.0493 |
| GP | 80.33 | $0.0186 \pm 0.0034$ | $0.0182 \pm 0.0019$ | $0.0186 \pm 0.0026$ | $0.0190 \pm 0.0022$ | $0.0164 \pm 0.0029$ | 0.0182 |
| I-Max | 80.20 | | | $0.0200 \pm 0.0033$ | | | |
| I-Max w. GP | 80.20 | | | **0.0121** $\pm 0.0048$ | | | |

Table A4: ECEs of post-hoc and during-training calibration. A WRN CIFAR100 classifier is trained in three modes: 1) no during-training calibration; 2) using entropy regularization (Pereyra et al., 2017); and 3) using Mixup data augmentation (Zhang et al., 2018; Thulasidasan et al., 2019). Taking each of the trained models as one baseline, we further perform post-hoc calibration. Note the best numbers per training mode is marked in bold and the underlined scores are the best across the three models.

| Post-Hoc Cal. | No Train Calibration | | Entr. Reg. | | Mixup | |
|---|---|---|---|---|---|---|
| | $_{CW}ECE_{cls-prior} \downarrow$ | $_{top1}ECE \downarrow$ | $_{CW}ECE_{cls-prior} \downarrow$ | $_{top1}ECE \downarrow$ | $_{CW}ECE_{cls-prior} \downarrow$ | $_{top1}ECE \downarrow$ |
| Baseline | 0.10434 | 0.06880 | 0.08860 | 0.04806 | 0.10518 | 0.04972 |
| Mtx Scal. w. $L_2$ | 0.10308 | 0.06560 | 0.08980 | 0.04650 | 0.10374 | 0.04852 |
| ETS-MnM | 0.09488 | 0.04820 | 0.09050 | 0.03900 | 0.09740 | 0.03676 |
| TS | 0.09436 | 0.05914 | 0.09376 | 0.05438 | 0.10282 | 0.04536 |
| GP | 0.10836 | 0.03360 | 0.10520 | 0.03728 | 0.10600 | 0.03514 |
| I-Max | **0.04574** | 0.01834 | **0.04712** | 0.02202 | **0.05534** | 0.02060 |
| I-Max w. TS | 0.05706 | 0.04342 | 0.04766 | 0.04478 | 0.06156 | 0.03264 |
| I-Max w. GP | 0.06130 | **0.01428** | 0.05114 | **0.01562** | 0.05992 | **0.01364** |

## A7 POST-HOC VS. DURING-TRAINING CALIBRATION

To calibrate a DNN-based classifier, there exists two groups of methods. One is to improve the calibration during training, whereas the other is post-hoc calibration. In this paper, we focus on post-hoc calibration because it is simple and does not require re-training of deployed models. In the following, we briefly discuss the advantages and disadvantages of post-hoc and during-training calibration.

In general, post-hoc and during-training calibration can be viewed as two orthogonal ways to improve the calibration, as they can be easily combined. Exemplarily, we compare/combine post-hoc calibration methods against/with during-training regularization which directly modifies the training objective to encourage less confident predictions through an entropy regularization term (Entr. Reg.) (Pereyra et al., 2017). Additionally, we adopt Mixup (Zhang et al., 2018) which is a data augmentation shown to improve calibration (Thulasidasan et al., 2019). We re-train the CIFAR100 WRN classifier respectively using Entr. Reg. and Mixup. It can be seen in Tab. A4 that compared to the Baseline model (without training calibration Entr. Reg. or Mixup), EntrReg improves the top1 ECE from 0.06880 to 0.04806. Further applying post-hoc calibration, I-Max and I-Max w. GP can reduce the 0.04806 to 0.02202 and 0.01562, respectively. This indicates that their combination is beneficial. In this particular case, we also observed that without Entr. Reg., directly post-hoc calibrating the Baseline model appears to be more effective, e.g., top 1 ECE of 0.01428 and class-wise ECE 0.04574. Switching to Mixup, the best top 1 ECE 0.01364 is attained by combining Mixup with post-hoc I-Max w. GP, while I-Max alone without during-training calibration is still the best at class-wise ECE.

While post-hoc calibrator is simple and effective at calibration, during-training techniques may deliver more than improving calibration, e.g., improving the generalization performance and providing robustness against adversarials. Therefore, instead of choosing either post-hoc or during training technique, we recommend the combination. While during-training techniques improve the generalization and robustness of the Baseline classifier, post-hoc calibration can further boost its calibration at a low computational cost.

## A8 TRAINING DETAILS

### A8.1 PRE-TRAINED CLASSIFICATION NETWORKS

We evaluate post-hoc calibration methods on four benchmark datasets, i.e., ImageNet Deng et al. (2009), CIFAR-100 Krizhevsky (2009), CIFAR-10 Krizhevsky (2009) and SVHN (Netzer et al., 2011), and across three modern DNNs for each dataset, i.e., InceptionResNetV2 Szegedy et al. (2017), DenseNet161 Huang et al. (2017) and ResNet152 He et al. (2016) for ImageNet, and Wide ResNet (WRN) Zagoruyko & Komodakis (2016) for the two CIFAR datasets and SVHN. Additionally, we

train DenseNet-BC ($L = 190$, $k = 40$) Huang et al. (2017) and ResNext8x64 Xie et al. (2017) for the two CIFAR datasets.

The ImageNet and CIFAR models are publicly available pre-trained networks and details are reported at the respective websites, i.e., ImageNet classifiers: `https://github.com/Cadene/pretrained-models.pytorch` and CIFAR classifiers: `https://github.com/bearpaw/pytorch-classification`.

### A8.2 Training Scaling Methods

The hyper-parameters were decided based on the original respective scaling methods publications with some exceptions. We found that the following parameters were the best for all the scaling methods. All scaling methods use the Adam optimizer with batch size 256 for CIFAR and 4096 for ImageNet. The learning rate was set to $10^{-3}$ for temperature scaling Guo et al. (2017) and Platt scaling Platt (1999), 0.0001 for vector scaling Guo et al. (2017) and $10^{-5}$ for matrix scaling Guo et al. (2017). Matrix scaling was further regularized as suggested by Kull et al. (2019) with a $L_2$ loss on the bias vector and the off-diagonal elements of the weighting matrix. BBQ Naeini et al. (2015), isotonic regression Zadrozny & Elkan (2002) and Beta Kull et al. (2017) hyper-parameters were taken directly from Wenger et al. (2020).

### A8.3 Training I-Max Binning

The I-Max bin optimization started from k-means++ initialization, which uses JSD instead of Euclidean metric as the distance measure, see Sec. A3.4. Then, we iteratively and alternatively updated $\{g_m\}$ and $\{\phi_m\}$ according to (5) until 200 iterations. With the attained bin edges $\{g_m\}$, we set the bin representatives $\{r_m\}$ based on the empirical frequency of class-1. If a scaling method is combined with binning, an alternative setting for $\{r_m\}$ is to take the averaged prediction probabilities based on the scaled logits of the samples per bin, e.g., in Tab. 2 in Sec. 4.2. Note that, for CW binning in  1, the number of samples from the minority class is too few, i.e., $25k/1k = 25$. We only have about $25/15 \approx 2$ samples per bin, which are too few to use empirical frequency estimates. Alternatively, we set $\{r_m\}$ based on the raw prediction probabilities. For ImageNet and CIFAR 10/100, which have test sets with uniform class priors, the used sCW setting is to share one binning scheme among all classes. Alternatively, for the imbalanced multi-class SVHN setting, we share binning among classes with similar class priors, and thus use the following class (i.e. digit) groupings: $\{0 - 1\}, \{2 - 4\}, \{5 - 9\}$.

## A9 Extend Tab. 1 for More Datasets and Models.

Tab. 1 in Sec. 4.1 of the main paper is replicated across datasets and models, where the basic setting remains the same. Specifically, three different ImageNet models can be found in Tab. A5, Tab. A6 and Tab. A7. Three models for CIFAR100 can be found in Tab. A8, Tab. A9 and Tab. A10. Similarly, CIFAR10 models can be found in Tab. A11, Tab. A12 and Tab. A13. The accuracy degradation of Eq. Mass reduces as the dataset has less number of classes, e.g., CIFAR10. This is a result of a higher class prior, where the one-vs-rest conversion becomes less critical for CIFAR10 than ImageNet. Nevertheless, its accuracy losses are still much larger than the other binning schemes, i.e., Eq. Size and I-Max binning. Therefore, its calibration performance is not considered for comparison. Overall, the observations of Tab. A5- A13 are similar to Tab. 1, showing the stable performance gains of I-Max binning across datasets and models.

## A10 Extend Tab. 2 for More Scaling Methods, Datasets and Models

Tab. 2 in Sec. 4.2 of the main paper is replicated across datasets and models, and include more scaling methods for comparison. The three binning methods all use the shared CW strategy, therefore 1k calibration samples are sufficient. The basic setting remains the same as Tab. 2. Three different ImageNet models can be found in Tab. A14, Tab. A15 and Tab. A16. Three models for CIFAR100

Table A5: Tab. 1 Extension: ImageNet - InceptionResNetV2

| Binn. | sCW(?) | size | $\text{Acc}_{\text{top1}}$ ↑ | $\text{Acc}_{\text{top5}}$ ↑ | $_{\text{cw}}\text{ECE}_{\frac{1}{K}}$ ↓ | $_{\text{top1}}\text{ECE}$ ↓ | NLL |
|---|---|---|---|---|---|---|---|
| Baseline | ✗ | - | **80.33** ± 0.15 | **95.10** ± 0.15 | 0.0486 ± 0.0003 | 0.0357 ± 0.0009 | 0.8406 ± 0.0095 |
| Eq. Mass | ✗ | 25k | 7.78 ± 0.15 | 27.92 ± 0.71 | 0.0016 ± 0.0001 | 0.0606 ± 0.0013 | 3.5960 ± 0.0137 |
| Eq. Mass | ✓ | 1k | 5.02 ± 0.13 | 26.75 ± 0.37 | 0.0022 ± 0.0001 | 0.0353 ± 0.0012 | 3.5272 ± 0.0142 |
| Eq. Size | ✗ | 25k | 78.52 ± 0.15 | 89.06 ± 0.13 | 0.1344 ± 0.0005 | 0.0547 ± 0.0017 | 1.5159 ± 0.0136 |
| Eq. Size | ✓ | 1k | 80.14 ± 0.23 | 88.99 ± 0.12 | 0.1525 ± 0.0023 | 0.0279 ± 0.0043 | 1.2671 ± 0.0130 |
| I-Max | ✗ | 25k | 80.27 ± 0.17 | 95.01 ± 0.19 | 0.0342 ± 0.0006 | 0.0329 ± 0.0010 | 0.8499 ± 0.0105 |
| I-Max | ✓ | 1k | 80.20 ± 0.18 | 94.86 ± 0.17 | **0.0302** ± 0.0041 | **0.0200** ± 0.0033 | **0.7860** ± 0.0208 |

Table A6: Tab. 1 Extension: ImageNet - DenseNet

| Binn. | sCW(?) | size | $\text{Acc}_{\text{top1}}$ ↑ | $\text{Acc}_{\text{top5}}$ ↑ | $_{\text{cw}}\text{ECE}_{\frac{1}{K}}$ ↓ | $_{\text{top1}}\text{ECE}$ ↓ | NLL |
|---|---|---|---|---|---|---|---|
| Baseline | ✗ | - | **77.21** ± 0.12 | **93.51** ± 0.14 | 0.0502 ± 0.0006 | 0.0571 ± 0.0014 | 0.9418 ± 0.0120 |
| Eq. Mass | ✗ | 25k | 18.48 ± 0.19 | 45.12 ± 0.26 | 0.0017 ± 0.0000 | 0.1657 ± 0.0020 | 2.9437 ± 0.0162 |
| Eq. Mass | ✓ | 1k | 17.21 ± 0.47 | 45.69 ± 1.22 | 0.0054 ± 0.0004 | 0.1572 ± 0.0047 | 2.9683 ± 0.0561 |
| Eq. Size | ✗ | 25k | 74.34 ± 0.28 | 88.27 ± 0.11 | 0.1272 ± 0.0011 | 0.0660 ± 0.0018 | 1.6699 ± 0.0165 |
| Eq. Size | ✓ | 1k | 77.06 ± 0.28 | 88.22 ± 0.10 | 0.1519 ± 0.0016 | 0.0230 ± 0.0050 | 1.3948 ± 0.0105 |
| I-Max | ✗ | 25k | 77.07 ± 0.13 | 93.40 ± 0.17 | 0.0334 ± 0.0004 | 0.0577 ± 0.0008 | 0.9492 ± 0.0130 |
| I-Max | ✓ | 1k | 77.13 ± 0.14 | 93.34 ± 0.17 | **0.0263** ± 0.0119 | **0.0201** ± 0.0088 | **0.9229** ± 0.0103 |

Table A7: Tab. 1 Extension: ImageNet - ResNet152

| Binn. | sCW(?) | size | $\text{Acc}_{\text{top1}}$ ↑ | $\text{Acc}_{\text{top5}}$ ↑ | $_{\text{cw}}\text{ECE}_{\frac{1}{K}}$ ↓ | $_{\text{top1}}\text{ECE}$ ↓ | NLL |
|---|---|---|---|---|---|---|---|
| Baseline | ✗ | - | **78.33** ± 0.17 | **94.00** ± 0.14 | 0.0500 ± 0.0004 | 0.0512 ± 0.0018 | 0.8760 ± 0.0133 |
| Eq. Mass | ✗ | 25k | 17.45 ± 0.10 | 44.87 ± 0.37 | 0.0017 ± 0.0000 | 0.1555 ± 0.0010 | 2.9526 ± 0.0168 |
| Eq. Mass | ✓ | 1k | 16.25 ± 0.54 | 45.53 ± 0.81 | 0.0064 ± 0.0004 | 0.1476 ± 0.0054 | 2.9471 ± 0.0556 |
| Eq. Size | ✗ | 25k | 75.50 ± 0.28 | 88.85 ± 0.19 | 0.1223 ± 0.0008 | 0.0604 ± 0.0017 | 1.6012 ± 0.0252 |
| Eq. Size | ✓ | 1k | 78.24 ± 0.16 | 88.81 ± 0.19 | 0.1480 ± 0.0015 | 0.0286 ± 0.0053 | 1.3308 ± 0.0178 |
| I-Max | ✗ | 25k | 78.24 ± 0.16 | 93.91 ± 0.17 | 0.0334 ± 0.0005 | 0.0521 ± 0.0015 | 0.8842 ± 0.0135 |
| I-Max | ✓ | 1k | 78.19 ± 0.21 | 93.82 ± 0.17 | **0.0295** ± 0.0030 | **0.0196** ± 0.0049 | **0.8638** ± 0.0135 |

Table A8: Tab. 1 Extension: CIFAR100 - WRN

| Binn. | sCW(?) | size | $\text{Acc}_{\text{top1}}$ ↑ | $_{\text{cw}}\text{ECE}_{\frac{1}{K}}$ ↓ | $_{\text{top1}}\text{ECE}$ ↓ | NLL |
|---|---|---|---|---|---|---|
| Baseline | ✗ | - | **81.35** ± 0.13 | 0.1113 ± 0.0010 | 0.0748 ± 0.0018 | 0.7816 ± 0.0076 |
| Eq. Mass | ✗ | 5k | 60.78 ± 0.62 | 0.0129 ± 0.0010 | 0.4538 ± 0.0074 | 1.1084 ± 0.0117 |
| Eq. Mass | ✓ | 1k | 62.04 ± 0.53 | 0.0252 ± 0.0032 | 0.4744 ± 0.0049 | 1.1789 ± 0.0308 |
| Eq. Size | ✗ | 5k | 80.39 ± 0.36 | 0.1143 ± 0.0013 | 0.0783 ± 0.0032 | 1.0772 ± 0.0184 |
| Eq. Size | ✓ | 1k | 81.12 ± 0.15 | 0.1229 ± 0.0030 | 0.0273 ± 0.0055 | 1.0165 ± 0.0105 |
| I-Max | ✗ | 5k | 81.22 ± 0.12 | 0.0692 ± 0.0020 | 0.0751 ± 0.0024 | 0.7878 ± 0.0090 |
| I-Max | ✓ | 1k | 81.30 ± 0.22 | **0.0518** ± 0.0036 | **0.0231** ± 0.0067 | **0.7593** ± 0.0085 |

Table A9: Tab. 1 Extension: CIFAR100 - ResNeXt8x64

| Binn. | sCW(?) | size | $\text{Acc}_{\text{top1}}$ ↑ | $_{\text{cw}}\text{ECE}_{\frac{1}{K}}$ ↓ | $_{\text{top1}}\text{ECE}$ ↓ | NLL |
|---|---|---|---|---|---|---|
| Baseline | ✗ | - | 81.93 ± 0.08 | 0.0979 ± 0.0015 | 0.0590 ± 0.0028 | 0.7271 ± 0.0026 |
| Eq. Mass | ✗ | 5k | 63.02 ± 0.54 | 0.0131 ± 0.0012 | 0.4764 ± 0.0057 | 1.0535 ± 0.0191 |
| Eq. Mass | ✓ | 1k | 64.48 ± 0.64 | 0.0265 ± 0.0011 | 0.4980 ± 0.0070 | 1.1232 ± 0.0277 |
| Eq. Size | ✗ | 5k | 80.81 ± 0.26 | 0.1070 ± 0.0008 | 0.0700 ± 0.0030 | 1.0178 ± 0.0066 |
| Eq. Size | ✓ | 1k | 81.99 ± 0.21 | 0.1195 ± 0.0013 | 0.0230 ± 0.0033 | 0.9556 ± 0.0071 |
| I-Max | ✗ | 5k | **81.99** ± 0.08 | 0.0601 ± 0.0027 | 0.0627 ± 0.0034 | 0.7318 ± 0.0026 |
| I-Max | ✓ | 1k | 81.96 ± 0.14 | **0.0549** ± 0.0081 | **0.0205** ± 0.0074 | **0.7127** ± 0.0040 |

Table A10: Tab. 1 Extension: CIFAR100 - DenseNet

| Binn. | sCW(?) | size | $Acc_{top1}$ ↑ | $_{CW}ECE_{\frac{1}{K}}$ ↓ | $_{top1}ECE$ ↓ | NLL |
|---|---|---|---|---|---|---|
| Baseline | ✗ | - | **82.36** ± 0.26 | 0.1223 ± 0.0008 | 0.0762 ± 0.0015 | 0.7542 ± 0.0143 |
| Eq. Mass | ✗ | 5k | 57.23 ± 0.50 | 0.0117 ± 0.0011 | 0.4173 ± 0.0051 | 1.1819 ± 0.0228 |
| Eq. Mass | ✓ | 1k | 58.11 ± 0.21 | 0.0233 ± 0.0005 | 0.4339 ± 0.0024 | 1.2049 ± 0.0405 |
| Eq. Size | ✗ | 5k | 81.35 ± 0.23 | 0.1108 ± 0.0017 | 0.0763 ± 0.0029 | 1.0207 ± 0.0183 |
| Eq. Size | ✓ | 1k | 82.22 ± 0.30 | 0.1192 ± 0.0024 | 0.0219 ± 0.0021 | 0.9482 ± 0.0137 |
| I-Max | ✗ | 5k | 82.35 ± 0.26 | 0.0740 ± 0.0007 | 0.0772 ± 0.0010 | 0.7618 ± 0.0145 |
| I-Max | ✓ | 1k | 82.32 ± 0.22 | **0.0546** ± 0.0122 | **0.0189** ± 0.0071 | **0.7022** ± 0.0124 |

Table A11: Tab. 1 Extension: CIFAR10 - WRN

| Binn. | sCW(?) | size | $Acc_{top1}$ ↑ | $_{CW}ECE_{\frac{1}{K}}$ ↓ | $_{top1}ECE$ ↓ | NLL |
|---|---|---|---|---|---|---|
| Baseline | ✗ | - | **96.12** ± 0.14 | 0.0457 ± 0.0011 | 0.0288 ± 0.0007 | 0.1682 ± 0.0062 |
| Eq. Mass | ✗ | 5k | 91.06 ± 0.54 | 0.0180 ± 0.0045 | 0.0794 ± 0.0066 | 0.2066 ± 0.0091 |
| Eq. Mass | ✓ | 1k | 91.24 ± 0.27 | 0.0212 ± 0.0009 | 0.0836 ± 0.0091 | 0.2252 ± 0.0220 |
| Eq. Size | ✗ | 5k | 96.04 ± 0.14 | 0.0344 ± 0.0008 | 0.0290 ± 0.0013 | 0.2231 ± 0.0074 |
| Eq. Size | ✓ | 1k | 96.04 ± 0.15 | 0.0278 ± 0.0021 | 0.0105 ± 0.0028 | 0.2744 ± 0.0812 |
| I-Max | ✗ | 5k | 96.10 ± 0.14 | 0.0329 ± 0.0011 | 0.0276 ± 0.0007 | 0.1704 ± 0.0067 |
| I-Max | ✓ | 1k | 96.06 ± 0.13 | **0.0304** ± 0.0012 | **0.0113** ± 0.0039 | **0.1595** ± 0.0604 |

Table A12: Tab. 1 Extension: CIFAR10 - ResNext8x64

| Binn. | sCW(?) | size | $Acc_{top1}$ ↑ | $_{CW}ECE_{\frac{1}{K}}$ ↓ | $_{top1}ECE$ ↓ | NLL |
|---|---|---|---|---|---|---|
| Baseline | ✗ | - | **96.30** ± 0.18 | 0.0485 ± 0.0014 | 0.0201 ± 0.0021 | **0.1247** ± 0.0058 |
| Eq. Mass | ✗ | 5k | 89.40 ± 0.55 | 0.0168 ± 0.0037 | 0.0589 ± 0.0052 | 0.2011 ± 0.0085 |
| Eq. Mass | ✓ | 1k | 89.85 ± 0.61 | 0.0269 ± 0.0051 | 0.0676 ± 0.0127 | 0.2208 ± 0.0172 |
| Eq. Size | ✗ | 5k | 96.30 ± 0.20 | 0.0274 ± 0.0013 | 0.0174 ± 0.0013 | 0.1613 ± 0.0101 |
| Eq. Size | ✓ | 1k | 96.17 ± 0.24 | 0.0288 ± 0.0039 | 0.0114 ± 0.0025 | 0.2495 ± 0.0571 |
| I-Max | ✗ | 5k | 96.26 ± 0.20 | **0.0240** ± 0.0020 | 0.0167 ± 0.0014 | 0.1264 ± 0.0066 |
| I-Max | ✓ | 1k | 96.22 ± 0.21 | 0.0254 ± 0.0030 | **0.0104** ± 0.0025 | 0.1397 ± 0.0276 |

Table A13: Tab. 1 Extension Dataset: CIFAR10 - DenseNet

| Binn. | sCW(?) | size | $Acc_{top1}$ ↑ | $_{CW}ECE_{\frac{1}{K}}$ ↓ | $_{top1}ECE$ ↓ | NLL |
|---|---|---|---|---|---|---|
| Baseline | ✗ | - | 96.65 ± 0.09 | 0.0404 ± 0.001 | 0.0253 ± 0.0009 | 0.1564 ± 0.0075 |
| Eq. Mass | ✓ | 1k | 88.80 ± 0.47 | 0.0233 ± 0.0024 | 0.0637 ± 0.0023 | 0.2694 ± 0.0274 |
| Eq. Mass | ✗ | 5k | 89.51 ± 0.36 | 0.0137 ± 0.0039 | 0.0657 ± 0.0041 | 0.2283 ± 0.0101 |
| Eq. Size | ✓ | 1k | 96.64 ± 0.22 | 0.0262 ± 0.0035 | 0.0101 ± 0.0035 | 0.2465 ± 0.0543 |
| Eq. Size | ✗ | 5k | 96.74 ± 0.07 | 0.0301 ± 0.0012 | 0.0242 ± 0.0013 | 0.1912 ± 0.0075 |
| I-Max | ✓ | 1k | 96.59 ± 0.32 | 0.0261 ± 0.0025 | 0.0098 ± 0.0027 | 0.1208 ± 0.0044 |
| I-Max | ✗ | 5k | 96.71 ± 0.09 | 0.0284 ± 0.0013 | 0.0233 ± 0.0009 | 0.1608 ± 0.0086 |

Table A14: Tab. 2 Extension: ImageNet - InceptionResnetV2

| Calibrator | $Acc_{top1}$ ↑ | $Acc_{top5}$ ↑ | $_{cw}ECE_{\frac{1}{K}}$ ↓ | $_{top1}ECE$ ↓ | NLL | Brier |
|---|---|---|---|---|---|---|
| Baseline | 80.33 ± 0.15 | 95.10 ± 0.15 | 0.0486 ± 0.0003 | 0.0357 ± 0.0009 | 0.8406 ± 0.0095 | 0.1115 ± 0.0007 |
| | | | 25k Calibration Samples | | | |
| BBQ (Naeini et al., 2015) | 53.89 ± 0.30 | 88.63 ± 0.22 | 0.0287 ± 0.0009 | 0.2689 ± 0.0033 | 1.7104 ± 0.0370 | 0.3273 ± 0.0016 |
| Beta Kull et al. (2017) | 80.47 ± 0.14 | 94.84 ± 0.15 | 0.0706 ± 0.0003 | 0.0346 ± 0.0022 | 0.9038 ± 0.0270 | 0.1174 ± 0.0010 |
| Isotonic Reg. Zadrozny & Elkan (2002) | 80.08 ± 0.19 | 93.46 ± 0.20 | 0.0644 ± 0.0014 | 0.0468 ± 0.0020 | 1.8375 ± 0.0587 | 0.1203 ± 0.0012 |
| Platt Platt (1999) | 80.48 ± 0.14 | 95.18 ± 0.12 | 0.0597 ± 0.0007 | 0.0775 ± 0.0015 | 0.8083 ± 0.0106 | 0.1205 ± 0.0010 |
| Vec Scal. Kull et al. (2019) | 80.53 ± 0.19 | 95.18 ± 0.16 | 0.0494 ± 0.0002 | 0.0300 ± 0.0010 | 0.8269 ± 0.0097 | 0.1106 ± 0.0007 |
| Mtx Scal. Kull et al. (2019) | **80.78** ± 0.18 | **95.38** ± 0.15 | 0.0508 ± 0.0003 | 0.0282 ± 0.0014 | 0.8042 ± 0.0100 | 0.1090 ± 0.0006 |
| BWS Ji et al. (2019) | 80.33 ± 0.16 | 95.10 ± 0.16 | 0.0561 ± 0.0008 | 0.044 ± 0.0019 | 0.8273 ± 0.0105 | 0.1129 ± 0.0009 |
| ETS-MnM Zhang et al. (2020) | 80.33 ± 0.16 | 95.10 ± 0.16 | 0.0479 ± 0.0004 | 0.0358 ± 0.0009 | 0.8426 ± 0.0097 | 0.1115 ± 0.0008 |
| | | | 1k Calibration Samples | | | |
| TS Guo et al. (2017) | 80.33 ± 0.16 | 95.10 ± 0.16 | 0.0559 ± 0.0015 | 0.0439 ± 0.0022 | 0.8293 ± 0.0107 | 0.1134 ± 0.0010 |
| GP Wenger et al. (2020) | 80.33 ± 0.15 | 95.11 ± 0.15 | 0.0485 ± 0.0035 | 0.0186 ± 0.0034 | **0.7556** ± 0.0118 | **0.1069** ± 0.0007 |
| Eq. Mass | 5.02 ± 0.13 | 26.75 ± 0.37 | 0.0022 ± 0.0001 | 0.0353 ± 0.0012 | 3.5272 ± 0.0142 | 0.0489 ± 0.0012 |
| Eq. Size | 80.14 ± 0.23 | 88.99 ± 0.12 | 0.1525 ± 0.0023 | 0.0279 ± 0.0043 | 1.2671 ± 0.0130 | 0.1115 ± 0.0011 |
| I-Max | 80.20 ± 0.18 | 94.86 ± 0.17 | 0.0302 ± 0.0041 | 0.0200 ± 0.0033 | 0.7860 ± 0.0208 | 0.1116 ± 0.0008 |
| Eq. Mass w. TS | 5.02 ± 0.13 | 26.87 ± 0.43 | 0.0023 ± 0.0001 | 0.0357 ± 0.0012 | 3.5454 ± 0.0222 | 0.0490 ± 0.0012 |
| Eq. Mass w. GP | 5.02 ± 0.13 | 26.87 ± 0.43 | 0.0022 ± 0.0001 | 0.0353 ± 0.0012 | 3.4778 ± 0.0217 | 0.0489 ± 0.0012 |
| Eq. Size w. TS | 80.26 ± 0.18 | 88.99 ± 0.12 | 0.1470 ± 0.0007 | 0.0391 ± 0.0038 | 1.2721 ± 0.0116 | 0.1136 ± 0.0012 |
| Eq. Size w. GP | 80.26 ± 0.18 | 88.99 ± 0.12 | 0.1508 ± 0.0021 | 0.0140 ± 0.0056 | 1.2661 ± 0.0121 | 0.1105 ± 0.0008 |
| I-Max w. TS | 80.20 ± 0.18 | 94.87 ± 0.19 | 0.0354 ± 0.0124 | 0.0402 ± 0.0019 | 0.8339 ± 0.0108 | 0.1142 ± 0.0009 |
| I-Max w. GP | 80.20 ± 0.18 | 94.87 ± 0.19 | **0.0300** ± 0.0041 | **0.0121** ± 0.0048 | 0.7787 ± 0.0102 | 0.1111 ± 0.0006 |

can be found in Tab. A17, Tab. A18 and Tab. A19. Similarly, CIFAR10 models can be found in Tab. A20, Tab. A21 and Tab. A22.

Being analogous to Tab. 2, we observe that in most cases matrix scaling performs the best at the accuracy, but fail to provide satisfactory calibration performance measured by ECEs, Brier scores and NLLs. Among the scaling methods, GP (Wenger et al., 2020) is the top performing one. Among the binning schemes, our proposal of I-Max binning outperforms Eq. Mass and Eq. Size at accuracies, ECEs, NLLs and Brier scores. The combination of I-Max binning with GP excels at the ECE performance. Note that, among all methods, Eq. Mass binning suffers from severe accuracy degradation after multi-class calibration. The reason behind Eq. Mass binning was discussed in Sec. 3.3 of the main paper. Given the poor accuracy, it is not in the scope of calibration performance comparison.

We also observe that GP performs better at NLL/Brier than the I-Max variants. GP is trained by directly optimizing the NLL as its loss. As a non-parametric Bayesian method, GP has larger model expressive capacity than binning. While achieving better NLL/Brier, it costs significantly more computational complexity and memory. In contrast, I-Max only relies on logic comparisons at test time. Among the binning schemes, I-Max w. GP achieves the best NLL/Brier across the datasets and models. It is noted that I-Max w. GP remains to be a binning scheme. So, the combination does not change the model capacity of I-Max. GP is only exploited during training to improve the optimization on I-Max's bin representatives. Besides the low complexity benefit, I-Max w. GP as a binning scheme does not suffer from the ECE underestimation issue of scaling methods such as GP.

We further note that as a cross entropy measure between two distributions, the NLL would be an ideal metric for calibration evaluation. However, *empirical* NLL and Brier favor high accuracy and high confident classifiers, as each sample only having one hard label essentially implies the maximum confidence on a single class. For the this reason, during training, the empirical NLL loss will keep pushing the prediction probability to one even after reaching $100\%$ training set accuracy. As a result, the trained classifier showed poor calibration performance at test time (Guo et al., 2017). In contrast to NLL/Brier, empirical ECEs use hard labels differently. The ground truth correctness associated to the prediction confidence $p$ is estimated by averaging over the hard labels of the samples receiving the prediction probability $p$ or close to $p$. Due to averaging, the empirical ground truth correctness is usually not a hard label. Lastly, we use a small example to show the difference between the NLL/Brier and ECE: for $N$ predictions, all assigned a confidence of $1.0$ and containing $M$ mistakes, the calibrated confidence is $M/N < 1$. Unlike ECE, the NLL/Brier loss is only non-zero only for the $M$ wrong predictions, despite all $N$ predictions being miscalibrated. This example shows that NLL/Brier penalize miscalibration far less than ECE.

Table A15: Tab. 2 Extension: ImageNet - DenseNet

| Calibrator | $Acc_{top1}$ ↑ | $Acc_{top5}$ ↑ | $_{CW}ECE_{\frac{1}{K}}$ ↓ | $_{top1}ECE$ ↓ | NLL | Brier |
|---|---|---|---|---|---|---|
| Baseline | 77.21 ± 0.12 | 93.51 ± 0.14 | 0.0502 ± 0.0006 | 0.0571 ± 0.0014 | 0.9418 ± 0.0120 | 0.1228 ± 0.0009 |
| | | | 25k Calibration Samples | | | |
| BBQ (Naeini et al., 2015) | 54.69 ± 0.42 | 86.55 ± 0.19 | 0.0274 ± 0.0007 | 0.2819 ± 0.0050 | 1.9805 ± 0.0500 | 0.3355 ± 0.0026 |
| Beta Kull et al. (2017) | 77.35 ± 0.22 | 93.34 ± 0.17 | 0.0494 ± 0.0008 | 0.0253 ± 0.0022 | 0.9768 ± 0.0254 | 0.1209 ± 0.0010 |
| Isotonic Reg. Zadrozny & Elkan (2002) | 76.81 ± 0.24 | 91.98 ± 0.17 | 0.0577 ± 0.0003 | 0.0490 ± 0.0021 | 1.9819 ± 0.0634 | 0.1281 ± 0.0012 |
| Platt Platt (1999) | 77.43 ± 0.21 | 93.64 ± 0.15 | 0.0448 ± 0.0010 | 0.0906 ± 0.0022 | 0.9168 ± 0.0139 | 0.1297 ± 0.0012 |
| Vec Scal. Kull et al. (2019) | 77.44 ± 0.20 | 93.62 ± 0.17 | 0.0492 ± 0.0006 | 0.0516 ± 0.0018 | 0.9276 ± 0.0134 | 0.1208 ± 0.0011 |
| Mtx Scal. Kull et al. (2019) | **77.56** ± 0.11 | **93.81** ± 0.15 | 0.0498 ± 0.0006 | 0.0491 ± 0.0015 | 0.9159 ± 0.0158 | 0.1202 ± 0.0016 |
| BWS Ji et al. (2019) | 77.21 ± 0.12 | 93.51 ± 0.14 | 0.0395 ± 0.0007 | 0.0301 ± 0.0012 | 0.9106 ± 0.0116 | 0.1197 ± 0.0008 |
| ETS-MnM Zhang et al. (2020) | 77.21 ± 0.12 | 93.51 ± 0.14 | 0.0357 ± 0.0008 | 0.0234 ± 0.0011 | 0.9188 ± 0.0103 | 0.1194 ± 0.0006 |
| | | | 1k Calibration Samples | | | |
| TS Guo et al. (2017) | 77.21 ± 0.12 | 93.51 ± 0.15 | 0.0375 ± 0.0007 | 0.0300 ± 0.0019 | 0.9116 ± 0.0110 | 0.1197 ± 0.0008 |
| GP Wenger et al. (2020) | 77.22 ± 0.12 | 93.51 ± 0.13 | 0.0394 ± 0.0037 | 0.0268 ± 0.0035 | **0.8914** ± 0.0120 | **0.1188** ± 0.0005 |
| Eq. Mass | 17.21 ± 0.47 | 45.69 ± 1.22 | 0.0054 ± 0.0004 | 0.1572 ± 0.0047 | 2.9683 ± 0.0561 | 0.1671 ± 0.0046 |
| Eq. Size | 77.06 ± 0.28 | 88.22 ± 0.10 | 0.1519 ± 0.0016 | 0.0230 ± 0.0050 | 1.3948 ± 0.0105 | 0.1206 ± 0.0013 |
| I-Max | 77.13 ± 0.14 | 93.34 ± 0.17 | 0.0263 ± 0.0119 | 0.0201 ± 0.0088 | 0.9229 ± 0.0103 | 0.1201 ± 0.0010 |
| Eq. Mass w. TS | 17.21 ± 0.47 | 45.73 ± 1.07 | 0.0054 ± 0.0004 | 0.1571 ± 0.0047 | 2.9104 ± 0.0482 | 0.1671 ± 0.0046 |
| Eq. Mass w. GP | 17.21 ± 0.47 | 45.71 ± 1.08 | 0.0054 ± 0.0004 | 0.1571 ± 0.0047 | 2.9090 ± 0.0485 | 0.1671 ± 0.0046 |
| Eq. Size w. TS | 77.19 ± 0.12 | 88.22 ± 0.10 | 0.1464 ± 0.0005 | 0.0241 ± 0.0032 | 1.3928 ± 0.0106 | 0.1201 ± 0.0008 |
| Eq. Size w. GP | 77.19 ± 0.12 | 88.22 ± 0.10 | 0.1527 ± 0.0007 | 0.0215 ± 0.0037 | 1.3944 ± 0.0094 | 0.1200 ± 0.0005 |
| I-Max w. TS | 77.13 ± 0.14 | 93.34 ± 0.17 | 0.0320 ± 0.0026 | 0.0245 ± 0.0024 | 0.9242 ± 0.0117 | 0.1201 ± 0.0007 |
| I-Max w. GP | 77.13 ± 0.14 | 93.34 ± 0.17 | **0.0258** ± 0.0100 | **0.0204** ± 0.0021 | 0.9200 ± 0.0124 | 0.1201 ± 0.0005 |

Table A16: Tab. 2 Extension: ImageNet - ResNet152

| Calibrator | $Acc_{top1}$ ↑ | $Acc_{top5}$ ↑ | $_{CW}ECE_{\frac{1}{K}}$ ↓ | $_{top1}ECE$ ↓ | NLL | Brier |
|---|---|---|---|---|---|---|
| Baseline | 78.33 ± 0.17 | 94.00 ± 0.14 | 0.05 ± 0.0004 | 0.0512 ± 0.0018 | 0.8760 ± 0.0133 | 0.1174 ± 0.0013 |
| | | | 25k Calibration Samples | | | |
| BBQ (Naeini et al., 2015) | 55.04 ± 0.26 | 87.15 ± 0.21 | 0.0278 ± 0.0004 | 0.2840 ± 0.0028 | 1.8490 ± 0.0474 | 0.3361 ± 0.0014 |
| Beta Kull et al. (2017) | 78.44 ± 0.16 | 93.71 ± 0.20 | 0.0507 ± 0.0012 | 0.0264 ± 0.0010 | 0.9365 ± 0.0249 | 0.1174 ± 0.0013 |
| Isotonic Reg. Zadrozny & Elkan (2002) | 77.97 ± 0.07 | 92.33 ± 0.32 | 0.0590 ± 0.0016 | 0.0486 ± 0.0027 | 1.9437 ± 0.1020 | 0.1248 ± 0.0015 |
| Platt Platt (1999) | 78.56 ± 0.15 | 94.06 ± 0.19 | 0.0458 ± 0.0009 | 0.0852 ± 0.0021 | 0.8557 ± 0.0159 | 0.1246 ± 0.0015 |
| Vec Scal. Kull et al. (2019) | **78.61** ± 0.21 | 94.12 ± 0.18 | 0.0490 ± 0.0003 | 0.0469 ± 0.0017 | 0.8625 ± 0.0143 | 0.1159 ± 0.0012 |
| Mtx Scal. Kull et al. (2019) | 78.54 ± 0.23 | **94.14** ± 0.22 | 0.0496 ± 0.0004 | 0.0443 ± 0.0026 | 0.8583 ± 0.0180 | 0.1160 ± 0.0016 |
| BWS Ji et al. (2019) | 78.33 ± 0.18 | 94.00 ± 0.15 | 0.0402 ± 0.0005 | 0.0277 ± 0.0019 | 0.8488 ± 0.0127 | 0.1147 ± 0.0012 |
| ETS-MnM Zhang et al. (2020) | 78.33 ± 0.18 | 94.00 ± 0.15 | 0.0366 ± 0.0007 | 0.0198 ± 0.0006 | 0.8609 ± 0.0117 | 0.1145 ± 0.0011 |
| | | | 1k Calibration Samples | | | |
| TS Guo et al. (2017) | 78.33 ± 0.18 | 94.00 ± 0.15 | 0.0378 ± 0.0007 | 0.0285 ± 0.0023 | 0.8505 ± 0.0126 | 0.1147 ± 0.0012 |
| GP Wenger et al. (2020) | 78.33 ± 0.17 | 94.00 ± 0.14 | 0.0403 ± 0.0021 | 0.0202 ± 0.0030 | **0.8366** ± 0.0118 | **0.1138** ± 0.0012 |
| Eq. Mass | 16.25 ± 0.54 | 45.53 ± 0.81 | 0.0064 ± 0.0004 | 0.1476 ± 0.0054 | 2.9471 ± 0.0556 | 0.1579 ± 0.0052 |
| Eq. Size | 78.24 ± 0.16 | 88.81 ± 0.19 | 0.1480 ± 0.0015 | 0.0286 ± 0.0053 | 1.3308 ± 0.0178 | 0.1167 ± 0.0011 |
| I-Max | 78.19 ± 0.21 | 93.82 ± 0.17 | 0.0295 ± 0.0030 | 0.0196 ± 0.0049 | 0.8638 ± 0.0135 | 0.1157 ± 0.0012 |
| Eq. Mass w. TS | 16.25 ± 0.54 | 45.54 ± 0.71 | 0.0064 ± 0.0004 | 0.1476 ± 0.0054 | 2.9024 ± 0.0401 | 0.1579 ± 0.0052 |
| Eq. Mass w. GP | 16.25 ± 0.54 | 45.52 ± 0.74 | 0.0064 ± 0.0004 | 0.1475 ± 0.0054 | 2.9021 ± 0.040 | 0.1579 ± 0.0052 |
| Eq. Size w. TS | 78.27 ± 0.17 | 88.81 ± 0.19 | 0.1428 ± 0.0007 | 0.0225 ± 0.0022 | 1.3286 ± 0.0171 | 0.1153 ± 0.0013 |
| Eq. Size w. GP | 78.27 ± 0.17 | 88.81 ± 0.19 | 0.1475 ± 0.0016 | 0.0138 ± 0.0049 | 1.330 ± 0.0171 | 0.1150 ± 0.0012 |
| I-Max w. TS | 78.19 ± 0.21 | 93.82 ± 0.17 | **0.0281** ± 0.0029 | 0.0219 ± 0.0016 | 0.8637 ± 0.0125 | 0.1152 ± 0.0015 |
| I-Max w. GP | 78.19 ± 0.21 | 93.82 ± 0.17 | 0.0296 ± 0.0029 | **0.0144** ± 0.0050 | 0.8602 ± 0.0127 | 0.1150 ± 0.0014 |

Table A17: Tab. 2 Extension: CIFAR100 - WRN

| Calibrator | $\text{Acc}_{\text{top1}}\uparrow$ | $\text{cwECE}_{\frac{1}{K}}\downarrow$ | $_{\text{top1}}\text{ECE}\downarrow$ | NLL | Brier |
|---|---|---|---|---|---|
| Baseline | $81.35\pm0.13$ | $0.1113\pm0.0010$ | $0.0748\pm0.0018$ | $0.7816\pm0.0076$ | $0.1082\pm0.0021$ |
| | 5k Calibration Samples | | | | |
| BBQ (Naeini et al., 2015) | $80.44\pm0.19$ | $0.0576\pm0.0018$ | $0.0672\pm0.0044$ | $1.7976\pm0.0443$ | $0.1297\pm0.0019$ |
| Beta Kull et al. (2017) | $81.44\pm0.17$ | $0.0952\pm0.0006$ | $0.0379\pm0.0027$ | $0.7624\pm0.0148$ | $0.1018\pm0.0016$ |
| Isotonic Reg. Zadrozny & Elkan (2002) | $81.25\pm0.27$ | $0.0597\pm0.0029$ | $0.0487\pm0.0040$ | $1.4015\pm0.0748$ | $0.1059\pm0.0013$ |
| Platt Platt (1999) | $81.35\pm0.12$ | $0.0827\pm0.0014$ | $0.0585\pm0.0038$ | $0.7491\pm0.0073$ | $0.1026\pm0.0017$ |
| Vec Scal. Kull et al. (2019) | $81.35\pm0.21$ | $0.1063\pm0.0013$ | $0.0687\pm0.0029$ | $0.7619\pm0.0064$ | $0.1055\pm0.0017$ |
| Mtx Scal. Kull et al. (2019) | $\mathbf{81.44}\pm0.20$ | $0.1085\pm0.0008$ | $0.0692\pm0.0033$ | $0.7531\pm0.0078$ | $0.1059\pm0.0019$ |
| BWS Ji et al. (2019) | $81.35\pm0.14$ | $0.1069\pm0.0009$ | $0.0451\pm0.0028$ | $0.737\pm0.0057$ | $0.1037\pm0.0017$ |
| ETS-MnM Zhang et al. (2020) | $81.35\pm0.14$ | $0.0976\pm0.0019$ | $0.0451\pm0.0027$ | $0.7695\pm0.0052$ | $0.1027\pm0.0020$ |
| | 1k Calibration Samples | | | | |
| TS Guo et al. (2017) | $81.35\pm0.14$ | $0.0911\pm0.0036$ | $0.0511\pm0.0059$ | $0.7527\pm0.0074$ | $0.1036\pm0.0025$ |
| GP Wenger et al. (2020) | $81.34\pm0.12$ | $0.1074\pm0.0043$ | $0.0358\pm0.0039$ | $\mathbf{0.6943}\pm0.0025$ | $\mathbf{0.0996}\pm0.0019$ |
| Eq. Mass | $62.04\pm0.53$ | $0.0252\pm0.0032$ | $0.4744\pm0.0049$ | $1.1789\pm0.0308$ | $0.4606\pm0.0034$ |
| Eq. Size | $81.12\pm0.15$ | $0.1229\pm0.0030$ | $0.0273\pm0.0055$ | $1.0165\pm0.0105$ | $0.1039\pm0.0017$ |
| I-Max | $81.30\pm0.22$ | $0.0518\pm0.0036$ | $0.0231\pm0.0067$ | $0.7593\pm0.0085$ | $0.1016\pm0.0018$ |
| Eq. Mass w. TS | $62.04\pm0.53$ | $0.0253\pm0.0034$ | $0.4764\pm0.0052$ | $1.0990\pm0.0184$ | $0.4624\pm0.0037$ |
| Eq. Mass w. GP | $62.04\pm0.53$ | $0.0252\pm0.0032$ | $0.4749\pm0.0051$ | $1.1110\pm0.0226$ | $0.4610\pm0.0036$ |
| Eq. Size w. TS | $81.31\pm0.15$ | $0.1197\pm0.0029$ | $0.0362\pm0.0065$ | $1.0106\pm0.0113$ | $0.1038\pm0.0026$ |
| Eq. Size w. GP | $81.31\pm0.15$ | $0.1205\pm0.0025$ | $0.0189\pm0.0054$ | $1.0161\pm0.0115$ | $0.1032\pm0.0020$ |
| I-Max w. TS | $81.34\pm0.20$ | $\mathbf{0.051}\pm0.0035$ | $0.0365\pm0.0067$ | $0.7716\pm0.0066$ | $0.1025\pm0.0021$ |
| I-Max w. GP | $81.34\pm0.20$ | $0.0559\pm0.0089$ | $\mathbf{0.0179}\pm0.0046$ | $0.7609\pm0.0080$ | $0.1014\pm0.0014$ |

Table A18: Tab. 2 Extension: CIFAR100 - ResNeXt

| Calibrator | $\text{Acc}_{\text{top1}}\uparrow$ | $\text{cwECE}_{\frac{1}{K}}\downarrow$ | $_{\text{top1}}\text{ECE}\downarrow$ | NLL | Brier |
|---|---|---|---|---|---|
| Baseline | $81.93\pm0.08$ | $0.0979\pm0.0015$ | $0.0590\pm0.0028$ | $0.7271\pm0.0026$ | $0.0984\pm0.0022$ |
| | 5k Calibration Samples | | | | |
| BBQ (Naeini et al., 2015) | $81.06\pm0.30$ | $0.0564\pm0.0013$ | $0.0608\pm0.0058$ | $1.6878\pm0.0546$ | $0.1176\pm0.0022$ |
| Beta Kull et al. (2017) | $82.19\pm0.31$ | $0.0918\pm0.0020$ | $0.0368\pm0.0047$ | $0.7095\pm0.0074$ | $0.0947\pm0.0024$ |
| Isotonic Reg. Zadrozny & Elkan (2002) | $81.89\pm0.19$ | $0.0619\pm0.0023$ | $0.0503\pm0.0036$ | $1.3015\pm0.0656$ | $0.0995\pm0.0018$ |
| Platt Platt (1999) | $82.28\pm0.21$ | $0.0790\pm0.0025$ | $0.0534\pm0.0047$ | $0.7050\pm0.0045$ | $0.0961\pm0.0026$ |
| Vec Scal. Kull et al. (2019) | $82.24\pm0.27$ | $0.0963\pm0.0013$ | $0.0572\pm0.0037$ | $0.7129\pm0.0053$ | $0.0973\pm0.0021$ |
| Mtx Scal. Kull et al. (2019) | $\mathbf{82.38}\pm0.17$ | $0.0970\pm0.0014$ | $0.0578\pm0.0040$ | $0.7042\pm0.0046$ | $0.0973\pm0.0023$ |
| BWS Ji et al. (2019) | $81.93\pm0.08$ | $0.1045\pm0.0015$ | $0.0448\pm0.0044$ | $0.6897\pm0.0031$ | $0.0969\pm0.0017$ |
| ETS-MnM Zhang et al. (2020) | $81.93\pm0.08$ | $0.0932\pm0.0020$ | $0.0460\pm0.001$ | $0.7284\pm0.0029$ | $0.0963\pm0.0022$ |
| | 1k Calibration Samples | | | | |
| TS Guo et al. (2017) | $81.93\pm0.08$ | $0.0864\pm0.0036$ | $0.0525\pm0.0057$ | $0.7163\pm0.0037$ | $0.0975\pm0.0020$ |
| GP Wenger et al. (2020) | $81.93\pm0.09$ | $0.1025\pm0.0037$ | $0.0345\pm0.0038$ | $\mathbf{0.6456}\pm0.0071$ | $\mathbf{0.0927}\pm0.0019$ |
| Eq. Mass | $64.48\pm0.64$ | $0.0265\pm0.0011$ | $0.4980\pm0.0070$ | $1.1232\pm0.0277$ | $0.4770\pm0.0051$ |
| Eq. Size | $81.99\pm0.21$ | $0.1195\pm0.0013$ | $0.0230\pm0.0033$ | $0.9556\pm0.0071$ | $0.0974\pm0.0014$ |
| I-Max | $81.96\pm0.14$ | $0.0549\pm0.0081$ | $0.0205\pm0.0074$ | $0.7127\pm0.0040$ | $0.0959\pm0.0018$ |
| Eq. Mass w. TS | $64.48\pm0.64$ | $0.0262\pm0.0013$ | $0.5003\pm0.0066$ | $1.0468\pm0.0228$ | $0.4793\pm0.0048$ |
| Eq. Mass w. GP | $64.48\pm0.64$ | $0.0264\pm0.0012$ | $0.4986\pm0.0066$ | $1.0555\pm0.0227$ | $0.4776\pm0.0048$ |
| Eq. Size w. TS | $81.94\pm0.09$ | $0.1179\pm0.0015$ | $0.0343\pm0.0029$ | $0.9498\pm0.0058$ | $0.0968\pm0.0022$ |
| Eq. Size w. GP | $81.94\pm0.09$ | $0.1177\pm0.0009$ | $0.0151\pm0.0029$ | $0.9561\pm0.0056$ | $0.0959\pm0.0018$ |
| I-Max w. TS | $81.96\pm0.14$ | $\mathbf{0.053}\pm0.0073$ | $0.0333\pm0.0023$ | $0.7286\pm0.0029$ | $0.0964\pm0.0019$ |
| I-Max w. GP | $81.96\pm0.14$ | $0.0532\pm0.0077$ | $\mathbf{0.0121}\pm0.0026$ | $0.7111\pm0.0024$ | $0.0950\pm0.0017$ |

Table A19: Tab. 2 Extension Dataset: CIFAR100 - DenseNet

| Calibrator | $Acc_{top1}$ ↑ | $_{cw}ECE_{\frac{1}{K}}$ ↓ | $_{top1}ECE$ ↓ | NLL | Brier |
|---|---|---|---|---|---|
| Baseline | $82.36 \pm 0.26$ | $0.1223 \pm 0.0008$ | $0.0762 \pm 0.0015$ | $0.7542 \pm 0.0143$ | $0.1041 \pm 0.0008$ |
| | | | 5k Calibration Samples | | |
| BBQ (Naeini et al., 2015) | $81.56 \pm 0.22$ | $0.0567 \pm 0.0020$ | $0.0635 \pm 0.0052$ | $1.5876 \pm 0.0914$ | $0.1216 \pm 0.0026$ |
| Beta Kull et al. (2017) | $82.39 \pm 0.28$ | $0.0953 \pm 0.0013$ | $0.0364 \pm 0.0034$ | $0.6935 \pm 0.0185$ | $0.0966 \pm 0.0008$ |
| Isotonic Reg. Zadrozny & Elkan (2002) | $82.05 \pm 0.26$ | $0.0591 \pm 0.0016$ | $0.0506 \pm 0.0025$ | $1.3030 \pm 0.1107$ | $0.1019 \pm 0.0014$ |
| Platt Platt (1999) | $82.34 \pm 0.28$ | $0.0866 \pm 0.0012$ | $0.0491 \pm 0.0012$ | $0.6835 \pm 0.0138$ | $0.0969 \pm 0.0015$ |
| Vec Scal. Kull et al. (2019) | $82.38 \pm 0.32$ | $0.1195 \pm 0.0005$ | $0.0711 \pm 0.0015$ | $0.7362 \pm 0.0173$ | $0.1028 \pm 0.0015$ |
| Mtx Scal. Kull et al. (2019) | $\mathbf{82.53} \pm 0.19$ | $0.1214 \pm 0.0006$ | $0.0733 \pm 0.0013$ | $0.7360 \pm 0.0153$ | $0.1025 \pm 0.0015$ |
| BWS Ji et al. (2019) | $82.36 \pm 0.27$ | $0.1028 \pm 0.0013$ | $0.0445 \pm 0.0021$ | $0.682 \pm 0.0125$ | $0.0975 \pm 0.0008$ |
| ETS-MnM Zhang et al. (2020) | $82.36 \pm 0.27$ | $0.1007 \pm 0.0016$ | $0.0387 \pm 0.0012$ | $0.6986 \pm 0.0111$ | $0.0969 \pm 0.0008$ |
| | | | 1k Calibration Samples | | |
| TS Guo et al. (2017) | $82.36 \pm 0.27$ | $0.0938 \pm 0.0017$ | $0.0447 \pm 0.0023$ | $0.6851 \pm 0.0115$ | $0.0976 \pm 0.0008$ |
| GP Wenger et al. (2020) | $82.35 \pm 0.27$ | $0.1021 \pm 0.0032$ | $0.0338 \pm 0.0011$ | $\mathbf{0.6536} \pm 0.0120$ | $\mathbf{0.0943} \pm 0.0007$ |
| Eq. Mass | $58.11 \pm 0.21$ | $0.0233 \pm 0.0005$ | $0.4339 \pm 0.0024$ | $1.2049 \pm 0.0405$ | $0.4317 \pm 0.0017$ |
| Eq. Size | $82.22 \pm 0.30$ | $0.1192 \pm 0.0024$ | $0.0219 \pm 0.0021$ | $0.9482 \pm 0.0137$ | $0.0997 \pm 0.0014$ |
| I-Max | $82.32 \pm 0.22$ | $0.0546 \pm 0.0122$ | $0.0189 \pm 0.0071$ | $0.7022 \pm 0.0124$ | $0.0967 \pm 0.0019$ |
| Eq. Mass w. TS | $58.11 \pm 0.21$ | $0.0233 \pm 0.0006$ | $0.4347 \pm 0.0024$ | $1.1483 \pm 0.0102$ | $0.4324 \pm 0.0017$ |
| Eq. Mass w. GP | $58.11 \pm 0.21$ | $0.0233 \pm 0.0005$ | $0.4342 \pm 0.0024$ | $1.1508 \pm 0.0099$ | $0.4319 \pm 0.0018$ |
| Eq. Size w. TS | $82.40 \pm 0.24$ | $0.1134 \pm 0.0014$ | $0.0245 \pm 0.0025$ | $0.9427 \pm 0.0137$ | $0.0986 \pm 0.0013$ |
| Eq. Size w. GP | $82.40 \pm 0.24$ | $0.1166 \pm 0.0021$ | $0.0126 \pm 0.0012$ | $0.9455 \pm 0.0142$ | $0.0985 \pm 0.0013$ |
| I-Max w. TS | $82.36 \pm 0.21$ | $\mathbf{0.048} \pm 0.0090$ | $0.0237 \pm 0.0009$ | $0.7040 \pm 0.0104$ | $0.0967 \pm 0.0010$ |
| I-Max w. GP | $82.36 \pm 0.21$ | $0.0535 \pm 0.0121$ | $\mathbf{0.0114} \pm 0.0025$ | $0.6988 \pm 0.0104$ | $0.0964 \pm 0.0010$ |

Table A20: Tab. 2 Extension: CIFAR10 - WRN

| Calibrator | $Acc_{top1}$ ↑ | $_{cw}ECE_{\frac{1}{K}}$ ↓ | $_{top1}ECE$ ↓ | NLL | Brier |
|---|---|---|---|---|---|
| Baseline | $96.12 \pm 0.14$ | $0.0457 \pm 0.0011$ | $0.0288 \pm 0.0007$ | $0.1682 \pm 0.0062$ | $0.0307 \pm 0.0008$ |
| | | | 5k Calibration Samples | | |
| BBQ (Naeini et al., 2015) | $95.98 \pm 0.15$ | $0.0290 \pm 0.0047$ | $0.0198 \pm 0.0044$ | $0.2054 \pm 0.0156$ | $0.0314 \pm 0.0005$ |
| Beta Kull et al. (2017) | $96.31 \pm 0.06$ | $0.0504 \pm 0.0015$ | $0.0208 \pm 0.0023$ | $0.1335 \pm 0.0039$ | $0.0271 \pm 0.0007$ |
| Isotonic Reg. Zadrozny & Elkan (2002) | $96.20 \pm 0.12$ | $0.0241 \pm 0.0021$ | $0.0138 \pm 0.0017$ | $0.1764 \pm 0.0241$ | $0.0273 \pm 0.0005$ |
| Platt Platt (1999) | $96.24 \pm 0.09$ | $0.0489 \pm 0.0011$ | $0.0177 \pm 0.0015$ | $0.1359 \pm 0.0039$ | $0.0270 \pm 0.0006$ |
| Vec Scal. Kull et al. (2019) | $\mathbf{96.27} \pm 0.11$ | $0.0449 \pm 0.0008$ | $0.0229 \pm 0.0008$ | $0.1437 \pm 0.0050$ | $0.0286 \pm 0.0007$ |
| Mtx Scal. Kull et al. (2019) | $96.20 \pm 0.10$ | $0.0444 \pm 0.0005$ | $0.0277 \pm 0.0007$ | $0.1625 \pm 0.0062$ | $0.0302 \pm 0.0008$ |
| BWS Ji et al. (2019) | $96.12 \pm 0.14$ | $0.0467 \pm 0.0012$ | $0.0195 \pm 0.0014$ | $0.1395 \pm 0.0077$ | $0.0279 \pm 0.0007$ |
| ETS-MnM Zhang et al. (2020) | $96.12 \pm 0.14$ | $0.0647 \pm 0.0014$ | $0.0329 \pm 0.0012$ | $0.1478 \pm 0.0038$ | $0.0270 \pm 0.0006$ |
| | | | 1k Calibration Samples | | |
| TS Guo et al. (2017) | $96.12 \pm 0.14$ | $0.0486 \pm 0.0024$ | $0.0205 \pm 0.0009$ | $0.1385 \pm 0.0048$ | $0.0278 \pm 0.0007$ |
| GP Wenger et al. (2020) | $96.10 \pm 0.13$ | $0.0549 \pm 0.0021$ | $0.0146 \pm 0.0022$ | $\mathbf{0.1281} \pm 0.0055$ | $0.0269 \pm 0.0009$ |
| Eq. Mass | $91.24 \pm 0.27$ | $0.0212 \pm 0.0009$ | $0.0836 \pm 0.0091$ | $0.2252 \pm 0.0220$ | $0.0858 \pm 0.0055$ |
| Eq. Size | $96.04 \pm 0.15$ | $0.0278 \pm 0.0021$ | $0.0105 \pm 0.0028$ | $0.2744 \pm 0.0812$ | $0.0305 \pm 0.0015$ |
| I-Max | $96.06 \pm 0.13$ | $0.0304 \pm 0.0012$ | $0.0113 \pm 0.0039$ | $0.1595 \pm 0.0604$ | $0.0274 \pm 0.0013$ |
| Eq. Mass w. TS | $91.24 \pm 0.27$ | $0.0219 \pm 0.0005$ | $0.0837 \pm 0.0092$ | $0.1944 \pm 0.0093$ | $0.0853 \pm 0.0054$ |
| Eq. Mass w. GP | $91.24 \pm 0.27$ | $0.0212 \pm 0.0008$ | $0.0821 \pm 0.0088$ | $0.1918 \pm 0.0091$ | $0.0851 \pm 0.0054$ |
| Eq. Size w. TS | $96.13 \pm 0.12$ | $0.0286 \pm 0.0018$ | $0.0125 \pm 0.0024$ | $0.1940 \pm 0.0063$ | $0.0296 \pm 0.0009$ |
| Eq. Size w. GP | $96.13 \pm 0.11$ | $\mathbf{0.0266} \pm 0.0016$ | $\mathbf{0.0066} \pm 0.0028$ | $0.1917 \pm 0.0058$ | $0.0292 \pm 0.0009$ |
| I-Max w. TS | $96.14 \pm 0.13$ | $0.0293 \pm 0.0010$ | $0.0163 \pm 0.0012$ | $0.1417 \pm 0.0047$ | $0.0280 \pm 0.0008$ |
| I-Max w. GP | $96.14 \pm 0.13$ | $0.0276 \pm 0.0011$ | $0.0074 \pm 0.0035$ | $0.1331 \pm 0.0042$ | $\mathbf{0.0268} \pm 0.0008$ |

Table A21: Tab. 2 Extension: CIFAR10 - ResNeXt

| Calibrator | Acc$_{top1}$ ↑ | $_{cw}$ECE$_{\frac{1}{K}}$ ↓ | $_{top1}$ECE ↓ | NLL | Brier |
|---|---|---|---|---|---|
| Baseline | 96.30 ± 0.18 | 0.0485 ± 0.0014 | 0.0201 ± 0.0021 | 0.1247 ± 0.0058 | 0.0266 ± 0.0013 |
| | | | 5k Calibration Samples | | |
| BBQ (Naeini et al., 2015) | 96.18 ± 0.12 | 0.0256 ± 0.0027 | 0.0166 ± 0.0020 | 0.1951 ± 0.0134 | 0.0286 ± 0.0004 |
| Beta Kull et al. (2017) | 96.31 ± 0.22 | 0.0517 ± 0.0011 | 0.0148 ± 0.0016 | 0.1163 ± 0.0040 | 0.0256 ± 0.0011 |
| Isotonic Reg. Zadrozny & Elkan (2002) | 96.35 ± 0.20 | 0.0241 ± 0.0016 | 0.0129 ± 0.0008 | 0.1686 ± 0.0099 | 0.0264 ± 0.0011 |
| Platt Platt (1999) | 96.34 ± 0.19 | 0.0511 ± 0.0008 | 0.0143 ± 0.0017 | 0.1159 ± 0.0042 | 0.0256 ± 0.0011 |
| Vec Scal. Kull et al. (2019) | **96.37** ± 0.19 | 0.0495 ± 0.0017 | 0.0161 ± 0.0017 | 0.1189 ± 0.0053 | 0.0258 ± 0.0013 |
| Mtx Scal. Kull et al. (2019) | 96.34 ± 0.21 | 0.0492 ± 0.0020 | 0.0187 ± 0.0020 | 0.1225 ± 0.0060 | 0.0263 ± 0.0014 |
| BWS Ji et al. (2019) | 96.3 ± 0.19 | 0.0514 ± 0.0013 | 0.015 ± 0.0008 | 0.1199 ± 0.0048 | 0.0257 ± 0.0012 |
| ETS-MnM Zhang et al. (2020) | 96.3 ± 0.19 | 0.0547 ± 0.0013 | 0.0159 ± 0.0027 | 0.1193 ± 0.0043 | 0.0257 ± 0.0011 |
| | | | 1k Calibration Samples | | |
| TS Guo et al. (2017) | 96.30 ± 0.19 | 0.0524 ± 0.0028 | 0.0150 ± 0.0009 | 0.1182 ± 0.0051 | 0.0257 ± 0.0012 |
| GP Wenger et al. (2020) | 96.31 ± 0.17 | 0.0529 ± 0.0017 | 0.0125 ± 0.0021 | **0.1176** ± 0.0051 | **0.0258** ± 0.0011 |
| Eq. Mass | 89.85 ± 0.61 | 0.0269 ± 0.0051 | 0.0676 ± 0.0127 | 0.2208 ± 0.0172 | 0.0841 ± 0.0042 |
| Eq. Size | 96.17 ± 0.24 | 0.0288 ± 0.0039 | 0.0114 ± 0.0025 | 0.2495 ± 0.0571 | 0.0277 ± 0.0008 |
| I-Max | 96.22 ± 0.21 | 0.0254 ± 0.0030 | 0.0104 ± 0.0025 | 0.1397 ± 0.0276 | 0.0265 ± 0.0012 |
| Eq. Mass w. TS | 89.85 ± 0.61 | 0.0269 ± 0.0054 | 0.0676 ± 0.0128 | 0.1966 ± 0.0104 | 0.0844 ± 0.0043 |
| Eq. Mass w. GP | 89.85 ± 0.61 | 0.0266 ± 0.0049 | 0.0669 ± 0.0126 | 0.1962 ± 0.0106 | 0.0841 ± 0.0043 |
| Eq. Size w. TS | 96.29 ± 0.18 | 0.0270 ± 0.0022 | 0.0062 ± 0.0024 | 0.1574 ± 0.0091 | 0.0264 ± 0.0013 |
| Eq. Size w. GP | 96.29 ± 0.18 | 0.0271 ± 0.0020 | 0.0063 ± 0.0030 | 0.1576 ± 0.0093 | 0.0264 ± 0.0012 |
| I-Max w. TS | 96.28 ± 0.19 | 0.0224 ± 0.0016 | 0.0053 ± 0.0024 | 0.1208 ± 0.0058 | 0.0259 ± 0.0012 |
| I-Max w. GP | 96.28 ± 0.19 | **0.0223** ± 0.0018 | **0.0052** ± 0.0029 | 0.1206 ± 0.0061 | 0.0259 ± 0.0012 |

Table A22: Tab. 2 Extension: CIFAR10 - DenseNet

| Calibrator | Acc$_{top1}$ ↑ | $_{cw}$ECE$_{\frac{1}{K}}$ ↓ | $_{top1}$ECE ↓ | NLL | Brier |
|---|---|---|---|---|---|
| Baseline | 96.65 ± 0.09 | 0.0404 ± 0.0010 | 0.0253 ± 0.0009 | 0.1564 ± 0.0075 | 0.0259 ± 0.0007 |
| | | | 5k Calibration Samples | | |
| BBQ (Naeini et al., 2015) | 96.75 ± 0.19 | 0.0245 ± 0.0030 | 0.0170 ± 0.0022 | 0.1806 ± 0.0105 | 0.0279 ± 0.0010 |
| Beta Kull et al. (2017) | 96.81 ± 0.10 | 0.0468 ± 0.0003 | 0.0154 ± 0.0013 | 0.1151 ± 0.0042 | 0.0234 ± 0.0007 |
| Isotonic Reg. Zadrozny & Elkan (2002) | 96.84 ± 0.08 | 0.0236 ± 0.0022 | 0.0140 ± 0.0024 | 0.1501 ± 0.0137 | 0.0241 ± 0.0007 |
| Platt Platt (1999) | 96.82 ± 0.11 | 0.0459 ± 0.0007 | 0.0141 ± 0.0010 | 0.1154 ± 0.0040 | 0.0233 ± 0.0007 |
| Vec Scal. Kull et al. (2019) | **96.84** ± 0.14 | 0.0413 ± 0.0014 | 0.0223 ± 0.0010 | 0.1373 ± 0.0077 | 0.0249 ± 0.0007 |
| Mtx Scal. Kull et al. (2019) | 96.73 ± 0.09 | 0.0402 ± 0.0017 | 0.0245 ± 0.0008 | 0.1531 ± 0.0081 | 0.0257 ± 0.0007 |
| BWS Ji et al. (2019) | 96.65 ± 0.10 | 0.0423 ± 0.0010 | 0.0188 ± 0.0016 | 0.1239 ± 0.0065 | 0.0239 ± 0.0006 |
| ETS-MnM Zhang et al. (2020) | 96.65 ± 0.10 | 0.0527 ± 0.0012 | 0.0212 ± 0.0012 | 0.1196 ± 0.0038 | 0.0230 ± 0.0007 |
| | | | 1k Calibration Samples | | |
| TS Guo et al. (2017) | 96.65 ± 0.10 | 0.0425 ± 0.0005 | 0.0169 ± 0.0010 | 0.1186 ± 0.0051 | 0.0237 ± 0.0006 |
| GP Wenger et al. (2020) | 96.66 ± 0.09 | 0.0490 ± 0.0022 | 0.0135 ± 0.0025 | **0.1143** ± 0.0048 | **0.0228** ± 0.0007 |
| Eq. Mass | 88.80 ± 0.47 | 0.0233 ± 0.0024 | 0.0637 ± 0.0023 | 0.2694 ± 0.0274 | 0.0881 ± 0.0033 |
| Eq. Size | 96.64 ± 0.22 | 0.0262 ± 0.0035 | 0.0101 ± 0.0035 | 0.2465 ± 0.0543 | 0.0256 ± 0.0003 |
| I-Max | 96.59 ± 0.32 | 0.0261 ± 0.0025 | 0.0098 ± 0.0027 | 0.1208 ± 0.0044 | 0.0239 ± 0.0005 |
| Eq. Mass w. TS | 88.80 ± 0.47 | 0.0234 ± 0.0026 | 0.0626 ± 0.0023 | 0.2102 ± 0.0051 | 0.0877 ± 0.0030 |
| Eq. Mass w. GP | 88.80 ± 0.47 | 0.0233 ± 0.0026 | 0.0634 ± 0.0025 | 0.2098 ± 0.0053 | 0.0880 ± 0.0030 |
| Eq. Size w. TS | 96.75 ± 0.10 | 0.0250 ± 0.0011 | 0.0133 ± 0.0014 | 0.1657 ± 0.0056 | 0.0249 ± 0.0007 |
| Eq. Size w. GP | 96.77 ± 0.10 | 0.0242 ± 0.0022 | 0.0050 ± 0.0012 | 0.1612 ± 0.0048 | 0.0245 ± 0.0005 |
| I-Max w. TS | 96.81 ± 0.15 | 0.0229 ± 0.0016 | 0.0125 ± 0.0017 | 0.1224 ± 0.0056 | 0.0239 ± 0.0007 |
| I-Max w. GP | 96.81 ± 0.15 | **0.0218** ± 0.0012 | **0.0048** ± 0.0009 | 0.1173 ± 0.0054 | 0.0231 ± 0.0005 |

