# OpenReview forum: "Multi-Class Uncertainty Calibration via Mutual Information Maximization-based Binning"
_ICLR.cc/2021/Conference — ICLR 2021 Poster_

### Official Review · AnonReviewer4 · 2020-10-28
**The paper provides enough motivation and intuition of why maximizing the mutual information between labels and quantized logits would help multi-class calibration, but there are some concerns that needs to be addressed.**

**Rating:** 5
**Confidence:** 4

**Review:**

This paper highlights the issues with the scaling method and histogram binning i.e., underestimate calibration error in scaling methods and failing to preserve classification accuracy, and sample-inefficiency in HB. They use the I-Max concept for binning, which maximizes the mutual information between labels and quantized logits. They claim that their approach mitigates potential loss in ranking performance and allows simultaneous improvement of ranking and calibration performance by disentangling the optimization of bin edges and representatives. They also propose a shared class-wise (sCW) strategy that fits a single calibrator on the merged training sets of all K class-wise problems to improve the  sample efficiency.

The paper is well written and the authors provide enough motivation and intuition of why maximizing the mutual information between labels and quantized logits would help multi-class calibration. There are some concerns and issues that I think needs to be addressed.

1- One approach in estimating uncertainty in classification is to choose a model and a regularized loss function to inherently learn a good representation. For example using confidence as a term for regularization in neural networks is proposed in Regularizing neural networks by penalizing confident output distributions (ICLR 2017) that penalizes low-entropy output distributions. I think it is worth comparing the results with such existing work and discussing the advantages and disadvantages since a similar concept has been used one while training the model and this paper as a post-hoc calibration.

2- It is interesting that a single calibrator on the merged training sets of all K class-wise problems (sCW) performs well. As it is mentioned in the paper, it introduces bias due to having samples drawn from the other classes. In HB, increasing its number of evaluation bins reduces the bias, but in (sCW) such bias can not be controlled. Moreover, Figure A2 shows it achieves smaller JSDs which is not expected. Is there any reason for that? What would happen if the number of bins is increased?


3- Based on the experimental results, it seems I-Max performs better than other binning approaches. However, compared to the scaling methods it seems GP (Wenger et al. 2020) performs better at NLL/Brier than the I-Max variants.

4- Even though the paper shows combining I-Max with GP improves the ECE, it is not clear how the issues of each approach will be handled. For example, the ECE might be underestimated.

---

> ### Author Response · Authors · 2020-11-21
> **Response to Reviewer 4**
>
>
> **R4-3)** *NLL/Brier Comparisons*
>
> Yes, we have observed that GP performs better than the I-Max binning variants in terms of NLL/Brier,
> and provided an explanation in Appendix A10 (formerly Appendix A9). As it may be too brief there,
> we elaborate more on this observation. GP is trained by directly optimizing the NLL as its loss. As
> a non-parametric Bayesian method, GP has a larger model expressive capacity than binning. While
> achieving better NLL/Brier, it costs significantly more computational complexity and memory. In
> contrast, I-Max only relies on logic comparisons at test time. Among the binning schemes, I-Max w.
> GP achieves the best NLL/Brier across the datasets and models. It is noted that I-Max w. GP remains
> to be a binning scheme. So, the combination does not change the model capacity of I-Max. GP is only
> exploited during training to improve the optimization of I-Max’s bin representatives. Besides the low
> complexity benefit, I-Max w. GP as a binning scheme does not suffer from the ECE underestimation
> issue of scaling methods such as GP.
>
> We further note that as a cross-entropy measure between two distributions, the NLL would be an ideal
> metric for calibration evaluation. However, "empirical" NLL and Brier favor high accuracy and high
> confident classifiers, as each sample only having one hard label essentially implies the maximum
> confidence on a single class. For this reason, during training, the empirical NLL loss will keep
> pushing the prediction probability to one even after reaching $100$% training set accuracy. As a result,
> the trained classifier showed poor calibration performance at test time (Guo et al., 2017). In contrast
> to NLL/Brier, empirical ECEs use hard labels differently. The ground truth correctness associated with
> the prediction confidence $p$ is estimated by averaging over the hard labels of the samples receiving
> the prediction probability $p$ or close to $p$. Due to averaging, the empirical ground truth correctness
> is usually not a hard label.
> Lastly, we use a small example to show the difference between the
> NLL/Brier and ECE: for $N$ predictions, all assigned a confidence of $1.0$ and containing $M$ mistakes,
> the calibrated confidence is $M/N < 1$. Unlike ECE, the NLL/Brier loss is only non-zero only for
> the $M$ wrong predictions, despite all $N$ predictions being miscalibrated. This example shows that
> NLL/Brier penalizes miscalibration far less than ECE.
> ***
> $ $
>
> **R4-4)** *Combination of I-Max with GP and How the issues of each approach will be handled.*
>
> The proposed combination of I-Max and GP aims at exploiting their
> complementary benefits. GP is a sample efficient scaling method for
> post-hoc calibration. However, it suffers from the issue of high
> complexity and ECE underestimation. On the contrary, I-Max has low
> complexity and its ECE estimation can converge to the ground truth ECE.
> To accurately set the bin representatives of the binning schemes, we often need a
> sufficient number of samples per bin. I-Max w. GP first sets the bin
> edges to bin the raw logits, which maximally preserve the label
> information. Then, it sets the bin representatives by averaging the
> GP-scaled prediction probabilities of the samples in each bin,
> exploiting the sample efficiency of GP. After setting the bin
> representatives, GP is no longer needed. At test time, I-Max w. GP
> remains to be a binning scheme, enjoying its low complexity and reliable
> ECE estimation. As a result of exploiting their complementary benefits
> (and in essence also addressing the underlying issues of each approach),
> we observe in Tab 2 that I-Max w. GP is the top-performing at ECE, being
> better than GP or I-Max alone.

---

> ### Author Response · Authors · 2020-11-21
> **Response to Reviewer 4**
>
> Thank you for your feedback. We hope that the following points can address your concerns and answer your questions.
>
> **R4-1)** *Post-hoc vs. during training calibration: Pros and Cons discussion.*
>
> Thank you for this suggestion. In general, we view post-hoc and during-training calibration as
> two orthogonal ways to improve the calibration, as they can be easily combined. For instance, we
> trained WRN CIFAR100 with the suggested entropy regularization (EntrReg) (Pereyra et al. 2017).
> Compared to the Baseline model (without EntrReg), it improves the top1 ECE from 0.06880 to
> 0.04806. Further applying post-hoc calibration, I-Max, and I-Max w. GP can reduce the 0.04806 to
> 0.02202 and 0.01562, respectively. This indicates that their combination is beneficial. In this particular
> case, we also observed that without EntrReg, directly post-hoc calibrating the Baseline model appears
> to be more effective, e.g., top 1 ECE of 0.01428. While post-hoc calibrator is simple and effective at
> calibration, during-training techniques may deliver more than improving calibration, e.g., improving
> the generalization performance and providing robustness against adversarials. Therefore, instead of
> choosing either post-hoc or during training technique, we recommend the combination. While during-
> training techniques improve the generalization and robustness of the Baseline classifier, post-hoc
> calibration can further boost its calibration at a low computational cost. We add the new experimental
> results to Appendix A7. In addition to EntrReg, we also experiment with data augmentation
> techniques that improve calibration during-training, e.g., Mixup (Zhang et. al. 2018, Thulasidasan
> et. al. 2019).
> ***
>
> $ $
>
> **R4-2)** *"In HB, increasing its number of evaluation bins reduces the bias, but in (sCW) such bias cannot be controlled. Moreover, Figure A2 shows it achieves smaller JSDs which is not expected. Is there any reason for that? What would happen if the number of bins is increased?"*
>
> We are not sure if we understand your questions correctly. So, before answering it, we would like to
> clarify that when using HB for calibration (e.g. I-Max and all its variants), the ECE evaluation does
> not require evaluation bins, see Fig 1-a) and the second paragraph in the introduction. If you refer to
> the use of Histogram Density Estimation (HDE) for evaluating the ECEs of scaling methods, it is true that increasing the number of
> evaluation bins reduces the bias, but the variance also increases. For a fixed number of evaluation
> samples, the optimal number of bins is unknown. This is a general issue of using scaling methods. In
> the paper, we use 100 evaluation bins as suggested by Wenger et al. 2020. Too few bins can result
> in underestimated ECEs. From Tab. A3, we also noticed that when using 100 evaluation bins, the ECE estimates are similar for different HDEs.
>
>
> The bias of sCW is different to and has no impact on the ECE estimation bias. As for controlling
> the bias of sCW, we can choose to share the binning schemes for classes belonging to the same
> class category and/or with similar class priors. However, at a small data regime, a one-for-all sharing
> strategy may still be favored, as the variance outweighs the bias. Here, we would also like to note
> that our main method I-Max is orthogonal to sCW. sCW is a strategy to construct the training set
> for calibrators (not only limited to binning-based calibrators, see Tab. 3). So, the number of bins is
> irrelevant to the bias of sCW.
>
> In Fig. A2, sCW does not always have smaller JSDs than CW, for instance, class 7 with the
> samples larger than 2k (the blue bar "sCW" is larger than the orange bar "CW"). So, for the class-7,
> the bias of merging logits starts outweighing the variance when the number of samples is more than
> 2k. Unfortunately, we don’t have more samples to further evaluate JSDs, i.e., making the variance
> sufficiently small to reveal the bias impact. Another reason that we don’t observe large JSDs of sCW
> for CIFAR10 is that the logit distributions of the 10 classes are similar. Therefore, the bias of sCW is
> small, making CIFAR10 a good use case of sCW. Note, for JSD evaluation, the histogram estimator
> sets the bin number as the maximum of ‘sturges’ and ‘fd’ estimators, both of them optimize their bin
> setting towards the number of samples.

---

### Official Review · AnonReviewer3 · 2020-10-29
**Interesting idea with extensive experiments**

**Rating:** 7
**Confidence:** 3

**Review:**

Update after the rebuttal: The authors have answered my concerns. I believe the paper should be accepted and would be a nice contribution to the current research.

===============================================================



The paper proposes a novel approach for post-hoc calibration of outputs of the neural networks to estimate uncertainty of its prediction. The paper considers the histogram binning approach (in contrast to scaling approaches existing in the literature) and utilises the information theory in building bins.

Strong points:
* The work is very well placed in the context of the existing literature identifying the current gaps
* Theoretically sound motivation of the approach
* Extensive empirical evaluation

Weak points:
* Some discussion of the cost of the proposed method is lacking - i.e. how much in terms of computational time and memory this new calibration method is?
* The methods are compared with respect to accuracy and Expected Calibration Error (ECE) only. It has been shown that ECE is not a good metric for comparing different methods (see, e.g. Ashukha, A., Lyzhov, A., Molchanov, D. and Vetrov, D., 2020. Pitfalls of in-domain uncertainty estimation and ensembling in deep learning. ICLR 2020).

I am recommending acceptance of the paper, though addressing the weak points above would largely improve the paper. The reasons for this decision is that strong points outweigh weak points: the proposed idea is interesting, it is shown that it is promising in practice (subject to not very good metrics) and the paper is mostly well written and easy to follow.

Questions to authors:
Could you please address raised weak points?

Additional feedback (not necessarily important for evaluation, but could help to improve the paper):
1. The part on shared class-wise binning is rather rushed in the main paper and it is not very clear. It is also rather independent contribution from the main I-Max calibration contribution. It would be better to somehow put them under one umbrella
2. Section 2. “Bayesian DNNs, e.g. (Blundell et al., 2015) and their approximations (Gal
& Ghahramani, 2016)” – a very arguable statement, I would suggest rephrasing it. Since Blundell et al. proposed variational inference which is also an approximation, and Gal & Ghahramani work is not an approximation of Blundell et al.’ model
3. Section 4.2. “Namely, matrix scaling w. L_2” dot after w is read as a full stop which is confusing
4. After eq.(5). "So, we can solve the problem by iteratively and alternately updating ${g_m}$ and ${\phi_m}$ based on (A12)." - it seems eq. 5 and A12 are the same and it would be more convenient to refer to eq. 5 in the text right after it.
5. I am a little bit missing the overall procedure of the proposed calibration. I.e. all details are there (especially if refer to appendix), but after reading the main paper, there is no feeling that I can now go and implement it for my problem. Maybe a pseudocode can help, or just step-by-step guidance

Minor:
Section 3, first paragraph: “Sec. 3.2)” – redundant bracket

---

> ### Author Response · Authors · 2020-11-21
> **Response to Reviewer 3**
>
> **R3-3)** *Additional feedback on presentation*
>
> Thank you very much for this detailed feedback of corrections and suggestions. These points are integrated into the revision. In particular, we revised Sec. 3.2 for a more detailed presentation of the class-wise calibration under the one-vs-rest strategy and the proposed sharing strategy, i.e., sCW. We would also like to point out Appendix A2 where we discuss the sCW contribution in more detail and with empirical analysis.
>
> Yes, the sCW contribution is orthogonal to I-Max. As we have shown in Tab. 3, it can also work with other methods based on one-vs-rest conversion, improving their calibration performance. We present both sCW and I-Max under the umbrella of class-wise calibration. sCW is a pragmatic technique for calibration set construction, addressing the class imbalance issue. As a binning scheme, I-Max is an outcome of maximizing the MI over the calibration set. sCW helps improve the sample efficiency at I-Max training.

---

> ### Author Response · Authors · 2020-11-21
> **Response to Reviewer 3**
>
> **R3-2)** *Pitfalls of ECE metric - Ashukha et al. 2020*
>
> In their paper, Ashukha et al. 2020 pointed out the pitfalls of using ECE for calibration performance evaluation. Thank you for suggesting this paper. We are already aware of some of these issues from Nixon et al. (cited by us and Ashukha et al. 2020) and therefore already considered these issues in our evaluation of the ECE. We added this paper to the related work in the revision. Although accurately evaluating calibration from a set of finite evaluation samples remains an open problem, ECE is arguably the most broadly used metric in the literature and we have tried to remedy the shortcomings in our evaluation. Here, we list the pitfalls mentioned by Ashukha et al. 2020 and individually explain how they are handled by our method and in our evaluation:
>
> 1. "Biased estimate of the true calibration": We would like to note that the empirical ECE
> estimation being biased or not also depends on the type of calibration methods. When
> applying binning for calibration, its empirical ECE estimation is asymptotically unbiased,
> namely, the empirical ECE can converge to the real ECE as the number of samples increases,
> see Fig. 1-b). This is also an important motivation for using binning for post-hoc calibration.
> On the other hand, when using scaling methods for calibration, their empirical ECE estimates
> are unfortunately not bias-free even with enough evaluation samples. See our discussion in
> the introduction (2nd paragraph).
>
> 2. "ECE-like scores cannot be optimized directly since they are minimized by a model with
> constant uniform predictions": Yes, ECE alone is not a proper scoring rule, but together
> with considering the accuracy loss the problem can be resolved. Namely, ECE alone can
> be minimized not only by a perfectly calibrated classifier that recovers the ground truth
> distribution but also by a naïve one that generates a uniform prediction for all inputs. The
> former is the right goal, whereas the latter is a cheating shortcut. To avoid rating a method
> that takes the shortcut highly, we always jointly examine the accuracy and ECE; the shortcut
> results in very poor accuracy. One important motivation of I-Max is to maximally preserve
> the label information, i.e., retaining the accuracy of the baseline classifier. Therefore, by
> design, I-Max pursues the first goal rather than the second shortcut.
>
> 3. "ECE only estimates miscalibration in terms of the maximum assigned probability whereas
> practical applications may require the full predicted probability vector to be calibrated.": As
> a proxy measure of how the full predicted probability vector is calibrated, class-wise ECEs
> are evaluated and reported together with top 1 ECE (the maximum assigned probability) in
> each experiment in our paper, therefore measuring the calibration of each class prediction.
>
> 4. "biases of ECE on different models may not be equal, rendering the miscalibration estimates
> incompatible.": As we are performing post-hoc calibration, we use the exact same baseline
> classifier for all post-hoc calibration methods. Therefore, this is not a critical concern in our
> study.
> $ $
>
> Ashukha et al. 2020 also suggested using an improved ECE metric from Nixon et al. 2019. We have
> also considered this in our evaluation:
>
> 1. Thresholding: similar to Nixon et al. 2019, we perform thresholding of the class-wise ECE
> metric to avoid the low probabilities to dominate the ECE metric. We also compared against
> multiple thresholds and the results are consistent across them.
>
> 2. Adaptive binning: For evaluating ECE of scaling methods, a common way is to estimate
> the distribution of their predictions using histogram density estimation (HDE). Nixon et
> al. 2019 suggested to configure HDE by equal mass instead of equal size binning, where
> equal mass is adaptive to the data distribution. As the ECE evaluation of scaling methods
> depends on the bin settings of HDE, the optimal choice for the bins is unclear. We designed
> 5 different binning schemes (including eq. mass) and reported the numbers in Tab. A3. In
> addition to HDE, we also evaluated the performance by KDE, being free from having to
> set the bins, instead, only picking the kernel (e.g., as suggested by J. Zhang et. al. 2020).
> Despite the ECEs of scaling methods varying over the 5 evaluation schemes,
> our proposal of I-Max has a constant ECE which outperforms all other methods across all
> ECE evaluation schemes. The reason why the ECE of I-Max is constant has been explained
> in the introduction. In short, its density estimation does not rely on HDE and KDE, so being
> independent of them as well as being free of the bias introduced by them at ECE estimation.

---

> > ### Comment · AnonReviewer3 · 2020-11-23
> > **Thank you for the detailed answer**
> >
> > Thank you for the detailed answer about ECE. Since the authors agree with ECE having issues, it may be a good idea to include other metrics for evaluation, e.g., calibrated log-likelihood suggested in Ashukha et al. 2020, or Patch Accuracy vs. Patch Uncertainty(PAvPU) (Mukhoti, J. and Gal, Y., 2018. Evaluating bayesian deep learning methods for semantic segmentation), or out-of-distribution detection, etc.

---

> > > ### Author Response · Authors · 2020-11-23
> > > **Clarity**
> > >
> > > To clarify, we think that the steps taken above to address the known pitfalls of ECE render it a useful and informative metric to compare and measure the calibration performance.
> > > However, we do agree that more metrics could be informative, so we will try to report the results on other metrics before the deadline.

---

> > > > ### Author Response · Authors · 2020-11-24
> > > > **Follow-up**
> > > >
> > > > Ashukha et al. 2020 suggested using the calibrated NLL as the poor calibration performance of a classifier may be easily improved by a simple post-hoc calibrator such as temperature scaling. To compare differently trained models, it was suggested to first apply a post-hoc calibration step before evaluating the NLL (i.e. calibrated NLL). Since we are comparing against different post-hoc calibration methods, their NLLs would be the so-called calibrated NLLs.  These NLL results have been shown in Tab. A14-A22 across all datasets and models, as well as the Brier scores.  We find that there is no single winner across all performance metrics, i.e., Accuracy, ECEs, and NLL/Brier.  Among the methods, only GP and I-Max are top-performing across all these metrics. GP consistently performs better at NLL/Brier, whereas I-Max is consistently better at ECEs.
> > > >
> > > > It is interesting to note that when a method achieves good Accuracy and ECEs, its NLL/Brier is also good. Though, the reverse does not hold true. For instance, matrix scaling with L2 regularization achieves good accuracy and NLL/Brier performance but performs poorly at ECEs. In our opinion, the empirical NLL/Brier as a calibration measure may be misleading. Calibration means that the predictive confidences shall match the distribution $p(y|x)$ (which is not a hard label). However, when a classifier predicts correctly, its empirical NLL/Brier loss is minimized by the confidence one (due to hard labels), rather than by the ground truth confidence $p(y|x)$) (which is not a hard label). So, empirical NLL/Brier favors over-confident correct predictions and under-confident wrong predictions. For example, an ambiguous but correctly-classified image should not have a confidence of 1, despite the given hard label indicating that.
> > > >
> > > > To perform a fair comparison on OOD, we also need baseline models which take into account OOD during training, e.g., Deep Anomaly Detection with Outlier Exposure from Hendrycks et al. ICLR2019 and Deep Verifier
> > > > Networks from Che et al. ICML2020.
> > > > Due to the time limit, we could not finish training them before the deadline.

---

> ### Author Response · Authors · 2020-11-21
> **Response to Reviewer 3**
>
> Thank you very much for the suggestions and the detailed feedback on improving the text of the
> paper. We hope the following three answers address your raised points.
>
>
> **R3-1)** *Cost of method (computational time/memory)*
>
> Thank you for the suggestion. We provide the pseudo-code of I-Max, i.e., Algorithm. 1, in the
> appendix, and analyze its complexity and memory cost in A3.6. Here we would like to point out
> that learning the I-Max bins and bin representatives are done within seconds for classifiers as large
> as ImageNet and performed purely in Numpy. More importantly, this complexity is not a test-time
> complexity. At test time, it becomes a matter of logic comparisons (i.e. finding the bin assignments
> for the logits and assigning them to bin representatives) and can calibrate an ImageNet test sample within
> a couple of microseconds. Memory-wise, we only need to store the bin edges and representatives (i.e.
> 2M − 1 floats, where M is the number of bins).

---

> > ### Comment · AnonReviewer3 · 2020-11-23
> > **Fantastic news**
> >
> > That's fantastic news about the proposed method. This big positive aspect of the method probably should be exploited more in the text, as I totally missed it after initial reading

---

> > > ### Author Response · Authors · 2020-11-23
> > > **Agreed**
> > >
> > > Yes, we will emphasize this point more in the text.
> > > In fact, this can be another very important advantage of I-Max over scaling methods such as GP which has a much higher training and test time complexity and memory cost.

---

### Official Review · AnonReviewer1 · 2020-10-30
**Elegant algorithm with great results, but why is it restricted to a uniform class prior?**

**Rating:** 7
**Confidence:** 4

**Review:**

Update: the authors went out of their way to address my concerns about the absence of the unbalanced class setting: they added a new datasets (SVHN), new results (table 4) and updated some of their explanations. All these additions seem satisfactory. I was also pleased with the feedback about computational cost (R3). I improved my rating.
While I agree with the concerns of reviewer 4 (those I could understand), they would apply to every publication I have read about calibration, and I think the authors addressed these concerns to the best of our current knowledge.

This paper proposes an information maximization binning scheme for calibration. Starting with a good introduction, a clear progression leads to the core algorithm described by theorem 1. Limits of previous histogram-based approaches, both in terms of performance or reliability of metric, are clearly demonstrated with clear figures and proper references.
While using information measures to drive histogram binning has been done, I assume that the current classification setting where one maximizes the MI between the logit and the class is novel (the authors do not give pointer to previous work here, only mentioning the Info Bottleneck without references).

Theorem 1 leads to an alternative minimization algorithm with analytical steps. I did not check the convergence behavior proof but Figure 3 is convincing enough. I did not fully understand the information bottleneck limit.

Experiments show first that the information binning strategy is far superior than equal-mass or size binning. Table 2 and 3 then shows how it improves on most scaling algorithms used for calibration. One detail I am not comfortable with: the ECE_{1/K} hack, as it looks like a last-minute addition to give even stronger gains to the I-MAX method. A more principled introduction would be better (see below).

This would be an excellent paper except for the following, which casts doubts whether all the steps of the method generalize to an unbalanced multiclass setting. It is probably possible to fix or explain before publication.
This paper relies on a very unnatural and unfortunate state-of-affair in ML: classes are equally distributed on the test data. The phrasing does even consider any other possibility, and some of the algorithms seem to be quite specific to this setup, requiring significant changes in the “real world” case where test classes are not equally distributed.

At the end of section 3.2, the authors propose an algorithm to merge {S_k} across K classes based on the observation that they have similar distribution. Rather than a proof, they run a simulation on ImageNet (Sec A.2) that shows it is better than binning each S_k separately. While the experiment is elegant, it probably strongly relies on the fact that each S_k has the same 1:K-1 split. What would happen if the classes follow a more realistic Zipf law, as observed in real NLP classification tasks? I would assume that the merging process could still be applicable, but applied to groups of {S_k} with similar class-0/class-1 distributions.

In section 4.1, the trick to remove from the measure of the ECE classes where the predicted probability is less than 1/K also depends on a uniform 1/K prior. It should also be adapted to a non-uniform prior.

---

> ### Author Response · Authors · 2020-11-21
> **Response to Reviewer 1**
>
> **R1-3)** *Prior work of using information measures, only mentioning the Info. bottleneck without references*
>
> Thank you for pointing the missing reference for information bottleneck (IB) proposed by Tishby et. al. 1999. It is added in the revision. From the perspective of using information measures to drive HB or, in a broader sense, quantization, our method falls into the framework of IB. Often, the design goal of quantization is to minimize the reconstruction loss or information loss before
> and after quantization. The IB framework instead considers a third variable which represents the relevant information carried by the variable before quantization. The goal then becomes to retain the information relevant to the third variable after quantization. Note that, IB as a theoretic framework is not application-oriented. In the context of multi-class calibration, the label is the third variable in addition to the logit and quantized logit. We care about how well the label can be predicted from the quantized logit rather than how well the logit can be reproduced after quantization. So, I-Max focuses on easing the difficulty of predicting the label from the quantized logits, rather than reconstructing the (input) logit from it. We have also identified that the bin edges of HB play the determinant role in preserving the label information. As two common choices in the literature for post-hoc
>  calibration, Eq. size, and Eq. mass HB were shown to be suboptimal. Our contribution lies in identifying and remedying the issues of the binning schemes at preserving the label information. In the revision, we add a section, i.e., Appendix A3.2, to discuss the connection with the IB.

---

> ### Author Response · Authors · 2020-11-21
> **Response to Reviewer 1**
>
> Thank you for your constructive comments which help improve our paper. We hope that our answers
> below can address your concerns.
>
> **R1-1)** *How the method generalizes to an imbalanced multi-class setting?*
>
> Thank you for pointing out that our experiments have only considered the balanced class setting. We note that the derivation of I-Max does not rely on the assumption of uniform class prior, but we agree that the shared binning strategy, i.e., sCW, could be more general and flexible than "one-for-all". As you mentioned, sharing among classes with similar class priors better suits the imbalanced multi-class setting. In fact, this strategy in the balanced multi-class setting boils down to "one-for-all".
>
> We also agree with the reviewer that an imbalanced multi-class setting is closer to reality. We are happy to add a new experiment based on SVNH whose classes are not equally distributed, i.e. the class priors ranging from 6% (e.g. digit 8) to 19% (e.g. digit 0). We reproduce Tab. 2 of the paper for the SVHN dataset, sharing I-Max binning among classes with similar class priors. Furthermore, in response to your comment on thresholding class-wise ECE and threshold being dependent on the class prior, we report class-wise ECE evaluated under the class prior ( CW-ECE_{cls-prior} ) as well as various other thresholds (discussed in more detail in the next question) in the new Tab. 4. We notice that despite this imbalance, our observations hold consistent and I-Max and its variants perform best compared to other calibrators. For instance, I-Max reduces the CW -ECE_{cls-prior} (and top1-ECE) of the Baseline from 0.0356 (0.0201) to 0.0245 (0.0164). Additionally, I-Max w. GP also reduces the CW ECE class-prior (and top1 ECE) of GP alone from 0.0341 (0.0104) to 0.0147 (0.0074). So, our method can generalize to the imbalanced multi-class setting.
>
> Thank you for the suggestion that improves our experiment setting.
>
> ***
> $ $
>
>
> **R1-2)** *Threshold adapts to the class prior, and more principled introduction of class-wise ECE thresholding*
>
> For setting the threshold, we meant to set it according to the class prior, which happens to be 1/K for a balanced multiclass setting. For the new SVHN dataset we investigate, the class prior does not coincide with 1/K, therefore, the threshold is adjusted accordingly. In the new Tab. 4 for SVHN, we add an ablation on setting various thresholds, namely, 1) 0 (no thresholding); 2) the class prior; 3) 1/K (any class with prediction probability below 1/K will not be the top-1); and 4) a relatively large
> number 0.5 (the case when the confidence on class-k outweighs NOT class-k). From Tab. 4, we can observe that I-Max and its variants are top-performing across the different thresholds.
>
> In the revision, we also improved the introduction of class-wise ECE thresholding, see Sec. 4 where we discuss ‘Training and Evaluation Details’ and explain the performance of Tab. 1 in Sec. 4.1. Here, we provide a brief justification for thresholding, as it is necessary rather than a "hack" for assessing the calibration performance of classes with small class priors. In Tab. 1, without thresholding, the class-wise ECE of the Baseline classifier InceptionResNetV2 trained on ImageNet is 0.000442. It is
> a very small number, thus may appear as the Baseline classifier is well calibrated. However, it is not the case, as we can observe from the top1 ECE (0.0357) which is still relatively large. The reason is as follows: when a class-k has a small class prior (e.g. 0.01 or 0.001), its empirical class-wise ECE score will be dominated by prediction samples where the class-k is not the ground truth. For
> these cases, a properly trained classifier, with acceptable accuracy, will often not rank the class-k among the top classes. In most cases, this will yield small calibration errors. While it is good to have many cases with small calibration errors, they should not wash out the calibration errors of the rest of the cases (prone to poor calibration) through performance averaging. These include cases where (1) the class-k is the ground truth class but incorrectly ranked and (2) the classifier misclassifies some class-j as class-k. The thresholding remedies the washing out by focusing more on crucial cases when evaluating the calibration performance of the class-k prediction (i.e. only averaging across cases where the prediction of the class-k is above a threshold). The need for thresholding when evaluating the class-wise ECE has also been discussed in Nixon et al. 2019. As for setting the threshold according to the class prior, it is an educated choice. Namely, the a-posteriori probability of the class-k from a proper classifier is an outcome of jointly considering the likelihood and the prior. If the a-posteriori probability of the class-k is lower than the prior after observing the sample, we believe that the class-k is unlikely to be the ground truth.

---

### Comment · ~David_Widmann1 · 2021-05-06
**A comment regarding the kernel calibration error (KCE)**

Congratulations to your paper, maximizing the mutual information between the bin indices and the targets seems to be a very neat idea!

I'd like to point out a possible misunderstanding about our work that you referred to in your paper. The kernel calibration error (KCE) that we [proposed for multi-class classification at NeurIPS 2019](http://papers.nips.cc/paper/9392-calibration-tests-in-multi-class-classification-a-unifying-framework) and that we [derived in a different more general way for arbitrary probabilistic predictive models at ICLR 2021](https://openreview.net/forum?id=-bxf89v3Nx) is fundamentally different from ECE and not based on kernel density estimation. One of the main advantages of KCE is that its estimation does not require to estimate the conditional distribution of the targets given a (set of) prediction(s) and therefore no binning or kernel density estimation is needed.

Our new general formulation highlights more clearly that KCE can be viewed as a version of the maximum mean discrepancy (MMD) applied to a special reformulation of the calibration setting. Other integral probability metrics, or possibly even more general distance measures, could be considered as well but we focused on the MMD since it leads to unbiased and consistent estimators (that do not require binning or kernel density estimation, and whose [convergence rate is independent of the number of dimensions of the joint space of predictions and targets](https://projecteuclid.org/journals/electronic-journal-of-statistics/volume-6/issue-none/On-the-empirical-estimation-of-integral-probability-metrics/10.1214/12-EJS722.full)) and also allows to perform statistical hypothesis tests of calibration based on different bounds and asymptotic approximations of the p-value, known from the literature of two-sample tests.

The [seminal work by Gretton et al.](https://www.jmlr.org/papers/volume13/gretton12a/gretton12a.pdf) explains the differences between kernel-density estimation approaches and the MMD (section 3.3.1), which apply in a similar way to the KCE. For instance, the authors note:
> An RKHS-based approach generalizes the L2 statistic [between Parzen windows density estimates] in a number of important respects. First,we may employ a much larger class of characteristic kernels that cannot be written as inner products between Parzen windows: several examples are given by Steinwart (2001, Section 3) and Micchelli et al. (2006, Section 3) (these kernels are universal, hence characteristic). We may further generalize to kernels on structured objects such as strings and graphs (Schölkopf et al., 2004), as done in our experiments (Section 8). Second, even when the kernel may be written as an inner product of Parzen windows on Rd, the D22 statistic with fixed bandwidth no longer converges to an L2 distance between probability density functions, hence it is more natural to define the statistic as an integral probability metric for a particular RKHS, as in Definition 2. Indeed, in our experiments, we obtain good performance in experimental settings where the dimensionality greatly exceeds the samplesize, and density estimates would perform very poorly (for instance the Gaussian toy example in Figure 5B, for which performance actually improves when the dimensionality increases; and the microarray data sets in Table 1). This suggests it is not necessary to solve the more difficult problem of density estimation in high dimensions to do two-sample testing.

---

### Decision · Program_Chairs · 2021-01-07
**Final Decision**

**Decision:**

Accept (Poster)

**Comment:**

The paper proposes to maximizing the mutual information to optimize the bin for multiclass calibration. The idea, technique, and presentation are good. The paper solves some multiclass calibration  issues. The author should revise the paper according the reviewer's comments before publish.